

# Global modelling of the total OH reactivity: investigations on the "missing" OH sink and its atmospheric implications

Valerio Ferracci[1,a], Ines Heimann[1], N. Luke Abraham[1,2], John A. Pyle[1,2] and Alexander T. Archibald[1,2]

[1]Centre for Atmospheric Science, Department of Chemistry, University of Cambridge, Lensfield Road, CB2 1EW, UK
[2]National Centre for Atmospheric Science, University of Cambridge, Cambridge, UK
[a]now at: Centre for Environmental and Agricultural Informatics, Cranfield University, College Road, MK43 0AL, UK

*Correspondence to*: Valerio Ferracci (vf257@cam.ac.uk)

**Abstract.** The hydroxyl radical (OH) plays a crucial role in the chemistry of the atmosphere as it initiates the removal of most trace gases. A number of field campaigns have observed the presence of a "missing" OH sink in a variety of regions across 10 the planet. Comparison of direct measurements of the OH loss frequency, also known as total OH reactivity ($k_{OH}$), with the sum of individual known OH sinks (obtained *via* the simultaneous detection of species such as volatile organic compounds and nitrogen oxides) indicates that, in some cases, up to 80 % of $k_{OH}$ is unaccounted for. In this work, the UM-UKCA chemistry-climate model was used to investigate the wider implications of the missing reactivity on the oxidising capacity of the atmosphere. Simulations of the present-day atmosphere were performed and the model was evaluated against an array of 15 field measurements to verify that the known OH sinks were reproduced well, with a resulting good agreement found for most species. Following this, an additional sink was introduced to simulate the missing OH reactivity as an emission of a hypothetical molecule, X, which undergoes rapid reaction with OH. The magnitude and spatial distribution of this sink were underpinned by observations of the missing reactivity. Model runs showed that the missing reactivity accounted for on average 6 % of the total OH loss flux at the surface, and up to 50 % in regions where emissions of the additional sink were high. The 20 lifetime of the hydroxyl radical was reduced by 3 % in the boundary layer, while tropospheric methane lifetime increased by 2 % when the additional OH sink was included. The UM-UKCA simulations also allowed us to establish the atmospheric implications of the newly characterised reactions of peroxy radicals ($RO_2$) with OH. While the effects of this chemistry on $k_{OH}$ were minor, the reaction of the simplest peroxy radical, $CH_3O_2$, with OH was found to be a major sink for $CH_3O_2$ and source of $HO_2$ over remote regions at the surface and in the free troposphere. Inclusion of this reaction in the model increased 25 tropospheric methane lifetime by up to 3 %, depending on its product branching. Simulations based on the latest kinetic and product information showed that this reaction cannot reconcile models with observations of atmospheric methanol, in contrast to recent suggestions.



# 1 Introduction

The removal of the vast majority of trace gases emitted into the atmosphere is initiated by reaction with the hydroxyl radical, OH. OH is primarily formed following the reaction of excited oxygen atoms, $O(^1D)$, originating from the photolysis of ozone at short wavelengths, with water:

$$O_3 + h\nu \ (\lambda < 310 \ nm) \quad \rightarrow \quad O(^1D) + O_2, \quad \text{(R1)}$$
$$O(^1D) + H_2O \quad \rightarrow \quad 2OH. \quad \text{(R2)}$$

OH generated *via* any route other than R1 and R2 is referred to as secondary OH; examples of processes yielding secondary
OH include the photolysis of $H_2O_2$ and the reaction of $HO_2$ with NO. Crucially, OH abundance and availability (and consequently the oxidising capacity of the atmosphere) are governed by the balance between OH sources (primary and secondary) and sinks, consisting of the totality of the species that react with OH: these include volatile organic compounds (VOCs), nitrogen oxides ($NO_x$) and many others species, both biogenic and anthropogenic.

In this respect, the total OH loss frequency, also known as the total OH reactivity ($k_{OH}$), is a useful measure of the total amount
of OH sinks present in a particular environment. $k_{OH}$ is defined as the pseudo-first order rate constant for OH loss and is equivalent to the inverse of the OH lifetime, $\tau_{OH}$, as shown in Eq. (1):

$$k_{OH} = \sum_{i=1}^{n} k_{OH+X_i}[X_i] = 1/\tau_{OH} , \quad \text{(1)}$$

where $[X_i]$ designates the concentration (usually in molecules $cm^{-3}$) of OH sink $X_i$, and $k_{OH+X_i}$ is the rate constant for the reaction of OH with $X_i$ (usually expressed in $cm^3$ $molecule^{-1}$ $s^{-1}$). It follows from Eq. (1) that, if the atmospheric abundance of all OH sinks is measured, and provided that the rate constants for their reaction with OH are known, $k_{OH}$ can be determined as the sum of the individual sink reactivities.

Over the last two decades, techniques capable of measuring the total OH reactivity *directly*, without the need to quantify
individual sinks, have become available: these rely either on direct measurements of the OH decay rate (Di Carlo et al., 2004; Ingham et al., 2009; Kovacs and Brune, 2001) or on the comparative reactivity method (Sinha et al., 2008). A review (Yang et al., 2016) recently described these techniques in detail, whilst the various instruments developed for direct measurements of $k_{OH}$ have been the subject of an extensive intercomparison (Fuchs et al., 2017). These techniques, when deployed in the field along with instruments for the detection of trace species, have enabled the comparison of direct measurements of the total $k_{OH}$
with the sum of reactivities of the individual OH sinks. In this respect, measurements of the total OH reactivity can be used to test our understanding of tropospheric oxidation: provided that the totality of OH sinks are accounted for and measured, the sum of individual reactivities and the total $k_{OH}$ should agree.





Field campaigns across the globe have however highlighted discrepancies between these two approaches, with most measurements reporting values of the total $k_{OH}$ higher than the sum of the individual reactivities (Yang et al., 2016). These results indicate that a fraction of the total OH reactivity cannot be accounted for; this is often referred to as "missing" reactivity and attributed to a "missing" OH sink. The magnitude of the observed missing reactivity in the literature varies depending on

the geographic location of the measurement and the season, but could amount to as much as 80% of the total $k_{OH}$ (Nölscher et al., 2016).

Many studies have attempted to identify the missing sink: while some authors have attributed the missing reactivity to the presence of primary emissions that escaped detection (Holzinger et al., 2005; Kaiser et al., 2016; Sinha et al., 2010), others have pointed at the reactions of OH with short-lived oxidation intermediates (Hansen et al., 2014; Nakashima et al., 2014),

which are notoriously challenging to measure in the field.

An exponential temperature dependence of the missing reactivity was observed during two campaigns carried out in the same North American forest, consistent with either primary biogenic emissions (Di Carlo et al., 2004) or with their oxidation products (Hansen et al., 2014). Some of the campaigns carried out in other forested environments also observed a similar trend (Kaiser et al., 2016; Mao et al., 2012; Zannoni et al., 2017), whereas others found no evidence for this correlation (Ren et al.,

2006b; Sinha et al., 2010).

In an attempt to account for the additional OH reactivity potentially arising from unmeasured oxidation intermediates, a number of studies invoked box modelling to determine the abundance of these species and their contribution to $k_{OH}$. These efforts have been met with mixed results: while some managed to reconcile the total $k_{OH}$ with the sum of reactivities once the oxidation intermediates were taken into account (Whalley et al., 2016), others obtained different degrees of improvement on the

agreement between the two, leaving different fractions of $k_{OH}$ still unaccounted for (Edwards et al., 2013; Elshorbany et al., 2012; Kaiser et al., 2016; Kovacs et al., 2003; Lee et al., 2009; Lou et al., 2010; Mao et al., 2012; Mogensen et al., 2011; Yang et al., 2017).

Regardless of its identity, the very presence of an additional OH sink would lead to shorter $\tau_{OH}$ in the real atmosphere than in current models; this would, in turn, lead to longer lifetimes for species that are primarily removed by reaction with OH, such

as the vast majority of biogenic and anthropogenic VOCs as well as high-impact greenhouse gases such as methane. Given the complex interactions of the OH radical in the photochemistry of the troposphere, global atmospheric modelling provides a powerful tool to investigate potential candidates for the missing sink, as well as to establish its impacts on the oxidising capacity of the lower atmosphere.

So far only two studies have attempted global modelling of $k_{OH}$: the focus of these works was either modelling the global OH

budget (Lelieveld et al., 2016) or that of the total reactive organic carbon (Safieddine et al., 2017). Detailed comparisons of the modelled $k_{OH}$ with observations or the missing reactivity were not addressed.

This work will make an extensive comparison between modelled $k_{OH}$ and observations, with particular attention to the contribution of individual sinks to the total OH reactivity. Section 3 describes our base integration and discusses a comparison with observations. Sections 4 and 5 tackle the challenge of modelling the missing reactivity using two approaches. Firstly



(section 4), we introduce an additional OH sink, the geographical distribution and abundance of which are underpinned by the observations of missing reactivity available. Secondly (Section 5), we include in the model the reactions of peroxy radicals ($RO_2$) with OH. As this novel $RO_2$ + OH chemistry has been characterised in the laboratory only in recent years, the role of $RO_2$ as an OH sink may have been overlooked (Fittschen et al., 2014). The implications of both approaches for the oxidising capacity of the atmosphere are then discussed.

## 2 Method

State-of-the-art chemistry-climate models have become an extremely important tool in the study of atmospheric science, allowing the exploration of a number of global scenarios with an unprecedented level of detail. However recent studies have shown that the way the chemistry is implemented in the model (*e.g.*, different oxidation schemes for complex emitted species) can have a major impact on crucial properties of the atmosphere such as the formation of tropospheric ozone (Squire et al., 2015). It is therefore important to validate these models against observations of relevant chemical species whenever possible. In this work, a global chemistry-climate model, the Met Office's Unified Model with the United Kingdom Chemistry and Aerosols scheme, UM-UKCA version 8.4, (Abraham et al., 2012) was used to investigate the total OH reactivity, $k_{OH}$. The model was run in the N96-L85 configuration, providing a horizontal resolution of 1.875° in longitude × 1.25° in latitude on 85 vertical levels from the surface up to a height of 85 km.

UM-UKCA was run with the Chemistry of the Stratosphere and Troposphere (CheST) scheme, combining previous tropospheric (O'Connor et al., 2014) and stratospheric (Morgenstern et al., 2009) chemical schemes as used by Banerjee et al. (2014). The scheme includes 83 advected chemical tracers and 310 photochemical reactions, describing the $O_x$, $HO_x$ and $NO_x$ chemical cycles and the oxidation of CO, methane, ethane, propane and isoprene (Archibald et al., 2010, 2011).

Reaction rate coefficients were based on recommended values from the International Union of Pure and Applied Chemistry (IUPAC) Subcommittee for Gas Kinetic Data Evaluation (http://www.iupac-kinetic.ch.cam.ac.uk), the JPL-NASA Evaluation of Chemical Kinetics and Photochemical Data for Use in Atmospheric Studies (Burkholder et al., 2015) and the Master Chemical Mechanism, MCM v3.2 (Jenkin et al., 2015) , *via* the website: http://mcm.leeds.ac.uk/MCM.

Surface emissions for the years 2000-2005 of nine trace gas species ($NO_x$, methane, CO, formaldehyde, ethane, propane, acetone, acetaldehyde and isoprene) were included based on Banerjee et al. (2014) as well as multi-level aircraft emissions for $NO_x$. Isoprene emissions were based on MEGAN (Guenther et al., 2006). The aerosol scheme also included emissions of organic carbon (OC, from both fossil fuels and biofuels), black carbon (BC, also from both fossil fuels and biofuels), monoterpenes, $SO_2$, dimethyl sulphide and biogenic methanol.

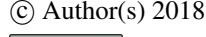



The model runs used in this work are described in Table 1. They comprise a Base run, discussed in detail in Section 3, a run with an imposed sink to account for the missing $k_{OH}$ ('X + OH run'), whose results are described in Section 4, and three additional experiments to explore the possible role of reactions of peroxy radicals. In each run, the model was run for five years, with one year spin-up time.

A number of diagnostics widely used in models to evaluate the oxidising capacity of the troposphere, such as methane lifetime with respect to tropospheric loss *via* reaction with OH ($\tau_{CH_4}$), OH lifetime ($\tau_{OH}$) and tropospheric ozone burden, were calculated for each model scenario. $\tau_{CH_4}$ was calculated in accordance with Lawrence et al. (2001), with the troposphere defined as the domain below 250 hPa. This is also consistent with the convention used in the Atmospheric Chemistry and Climate Model

10 Intercomparison Project (ACCMIP) (Naik et al., 2013; Voulgarakis et al., 2013). The same definition of the troposphere was used here in the calculation of tropospheric reaction fluxes and $\tau_{OH}$. For the calculation of the tropospheric ozone burden, the troposphere was defined as the domain in which the ozone mixing ratio was below 150 ppbv (or nmol/mol), in accordance with previous studies and model intercomparisons (Ehhalt et al., 2001; Stevenson et al., 2006; Young et al., 2013). Species lifetimes were calculated by dividing the species burden by their removal rate.

**Table 1: Summary of the model runs described in this work**

| Run name | Chemistry scheme | Description |
|---|---|---|
| Base run | CheST | This run provides a means to assess how well the model captures known OH sinks, *i.e.* the individual reactivities in the sum term of Eq. (1). It also provides a point of comparison for the runs that follow. The Base run is discussed in Section 3. |
| X + OH run | CheST with R3 | An additional OH sink, species X, is introduced in the model to account for the missing $k_{OH}$. This run is described in Section 4. |
| CH$_3$O$_2$ + OH run 1 | CheST with R4 | The multi-channel reaction of methyl peroxy radicals (CH$_3$O$_2$) with OH was included in the chemistry scheme with branching ratios $\alpha = 1$, $\gamma = 0$ [a]. This run is described in Section 5.2. |
| CH$_3$O$_2$ + OH run 2 | CheST with R4 | Same as CH$_3$O$_2$ + OH run 1 but with branching ratios $\alpha = 0.8$, $\gamma = 0.2$ [a]. This run is described in Section 5.2. |
| CH$_3$O$_2$ + OH run 3 | CheST with R4 | Same as CH$_3$O$_2$ + OH run 1 but with branching ratios $\alpha = 0.6$, $\gamma = 0.4$ [a]. This run is described in Section 5.2. |

**Notes**:

[a] branching ratios $\alpha$ and $\gamma$ are defined in Section 5.2



## 3 Comparison of modelled $k_{OH}$ and known OH sinks with observations

The modelled $k_{OH}$ at the surface, obtained from the Base run using the standard CheST scheme, is shown in Figure 1. The total OH reactivity is lowest over oceans and remote deserts ($< 1$ s$^{-1}$), highest over tropical forests ($> 10$ s$^{-1}$) and somewhat intermediate between these values in urban influenced areas. This global distribution and magnitude of $k_{OH}$ are in reasonably

good agreement with those calculated in previous modelling studies (Lelieveld et al., 2016; Safieddine et al., 2017).

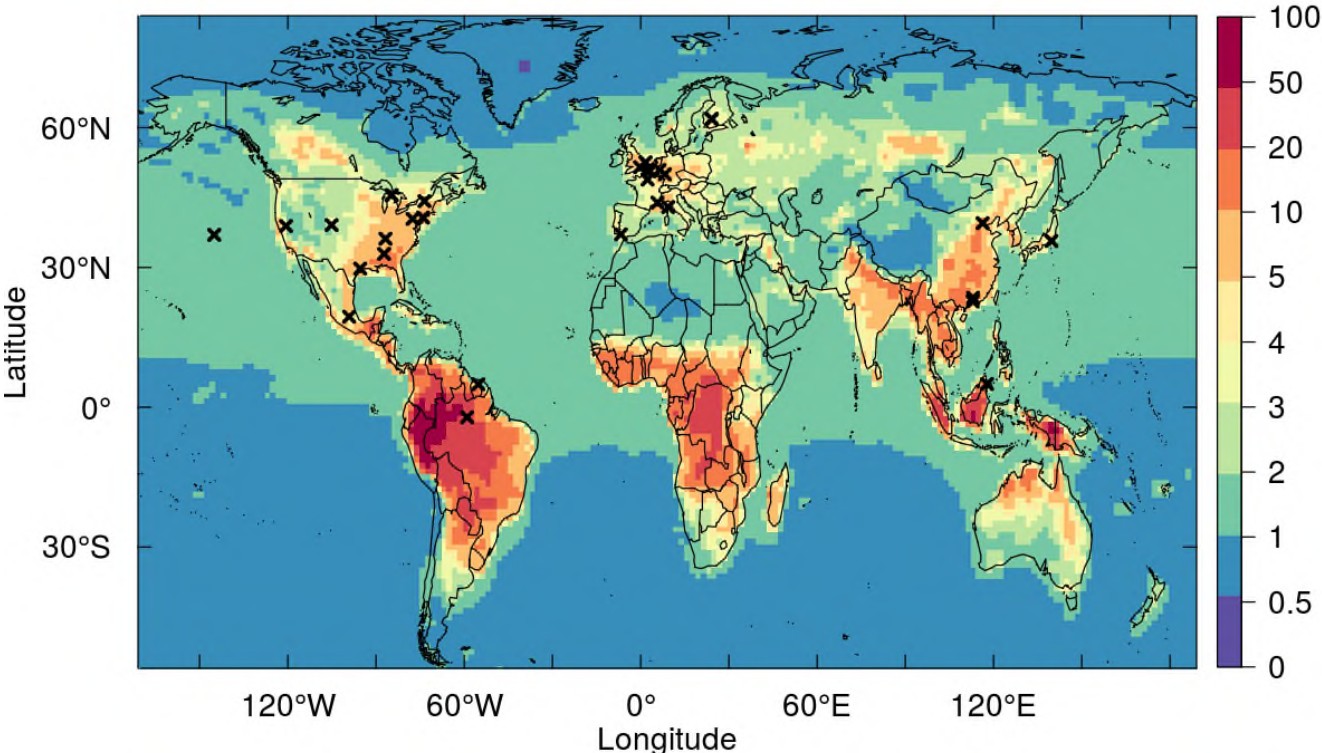

**Figure 1: Annual mean of the total OH reactivity (in s$^{-1}$) calculated in the Base run at the surface. Crosses indicate sites of field campaigns against which the model is compared.**

As described in Section 1, many measurements of $k_{OH}$ have been performed over the last two decades, with an exhaustive summary presented in a recent review (Yang et al., 2016). Only measurements of $k_{OH}$ and its speciation performed over reasonably long timescales ($\geq 1$ week) and covering the full diurnal variation of $k_{OH}$ and the OH sinks were considered in this work, in order to minimise biases due to day-to-day variability and to obtain a meaningful comparison with the model. A small number of field campaigns measured only the total $k_{OH}$ and not the abundance of the individual sinks, therefore

precluding the quantification of the missing reactivity or any further analysis on the speciation of $k_{OH}$ (Michoud et al., 2015; Ren et al., 2005; Sinha et al., 2008, 2012). For these reasons, 27 field measurements of the total $k_{OH}$ and of the individual OH sinks were used in the analysis described in this work; these are summarised in Table 2, where the values of the total $k_{OH}$ and




of the missing reactivity, averaged over the whole duration of each campaign, are reported. These observations were performed in a variety of environments, the vast majority of which were situated in the Northern Hemisphere (as shown in Figure 1). Measurement sites can be grouped into suburban (6 measurements), urban (10 measurements) and remote areas (11 measurements).

**Table 2: Total observed OH reactivity and missing reactivity from field campaigns. These values represent averages over the whole duration of each campaign. Measurement sites are grouped into three categories (suburban, urban, remote environments respectively).**

| | Location (Campaign) | Total observed $k_{OH}/s^{-1}$ * | Missing $k_{OH}/s^{-1}$ ** | Missing $k_{OH}$ after the inclusion of model intermediates/ $s^{-1}$ *** | Reference |
|---|---|---|---|---|---|
| **Suburban** | Whiteface Mountain, USA (PMTACS-NY2002) | 5.40 | 0.02 | | (Ren et al., 2006b) |
| | Weybourne, UK (TORCH-2) | 4.6 | 2.0 | 1.3 | (Lee et al., 2009) |
| | Yufa, China (CAREBeijing-2006) | 19.7 | 2.2 | | (Lu et al., 2010) |
| | Backgarden, China (PRIDE-PRD) | 31.4 | 15.7 | 6.3 | (Lou et al., 2010) |
| | Jülich, Germany (HOx Comp) | 8.6 | 3.2 | 2.5 | (Elshorbany et al., 2012) |
| | Heshan, China | 30.6 | 9.8 | 5.3 | (Yang et al., 2017) |
| | Ersa, Corsica (CARBOSOR-ChArMeX) | 5.6 | 2.3 | | (Zannoni et al., 2017) |
| **Urban** | Nashville, USA (SOS) | 11.0 | 3.8 | | (Kovacs et al., 2003) |
| | New York, USA (PMTACS-NY2001) | 18.8 | 0.7 | | (Ren et al., 2003a, 2003b) |
| | New York, USA (PMTACS-NY2004) | 25.1 | 4.0 | | (Ren et al., 2006a) |
| | Mexico City, Mexico (MCMA-2003) | 47.5 | 14.3 | | (Shirley et al., 2006) |
| | Houston, USA (TexAQS) | 9.4 | 0.4 | | (Mao et al., 2010) |
| | Houston, USA (TRAMP2006) | 12.24 | 0.03 | | (Mao et al., 2010) |
| | Paris, France (MEGAPOLI) | 40.3 | 22.8 | | (Dolgorouky et al., 2012) |
| | Lille, France | 7.4 | 0 | | (Hansen et al., 2015) |
| | London, UK (ClearfLo) | 18.1 | 5.9 | 2.7[†] | (Whalley et al., 2016) |
| **Remote** | Michigan, USA (Prophet2000) | 7.8 | 2.6 | | (Di Carlo et al., 2004) |
| | Hyytiälä, Finland (BFORM) | 8.6 | 3.9 | | (Sinha et al., 2010) |
| | Hyytiälä, Finland (HUMPPA-COPEC2010) | 11.5 | 8.9 | | (Nölscher et al., 2012) |
| | Rocky Mountains, USA (BEACHON-SRM08) | 6.7 | 2.1 | | (Nakashima et al., 2014) |
| | Michigan, USA (CABINEX) | 11.6 | 6.3 | | (Hansen et al., 2014) |
| | Amazon, Brazil (ATTO) dry season | 49.6 | 35.8 | | (Nölscher et al., 2016) |

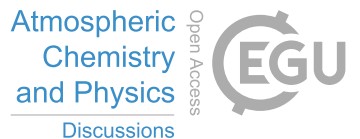

| | | | | |
|---|---|---|---|---|
| Amazon, Brazil (ATTO) wet season | 8.3 | 3.9 | | (Nölscher et al., 2016) |
| Haute Provence, France (CANOPEE) | 17.9 | 1.1 | | (Zannoni et al., 2016) |
| Borneo, Malaysia (OP3) | 15.3 | 10.2 | 3.9 | (Edwards et al., 2013) |
| Alabama, USA (SOAS) | 19.4 | 4.9 | 3.6 | (Kaiser et al., 2016) |
| California, USA (BEARPEX09) | 17.3 | 7.1 | 3.5 | (Mao et al., 2012) |
| North Pacific (INTEX-B) | 4.0 | 2.2 | | (Mao et al., 2009) |

**Notes:**

\* these values are the mean of the total $k_{OH}$ measured over the whole duration of each field campaign

\*\* missing reactivity calculated as the difference between the total $k_{OH}$ and the sum of the individual reactivities arising *exclusively* from measured OH sinks

\*\*\* missing reactivity calculated as the difference between the total $k_{OH}$ and the sum of the individual reactivities arising from *both* measured OH sinks *and* intermediates modelled in the particular studies referenced

[†]the addition of unidentified compounds observed by GC×GC-FID reduced the missing reactivity further to only ~1.1 s$^{-1}$

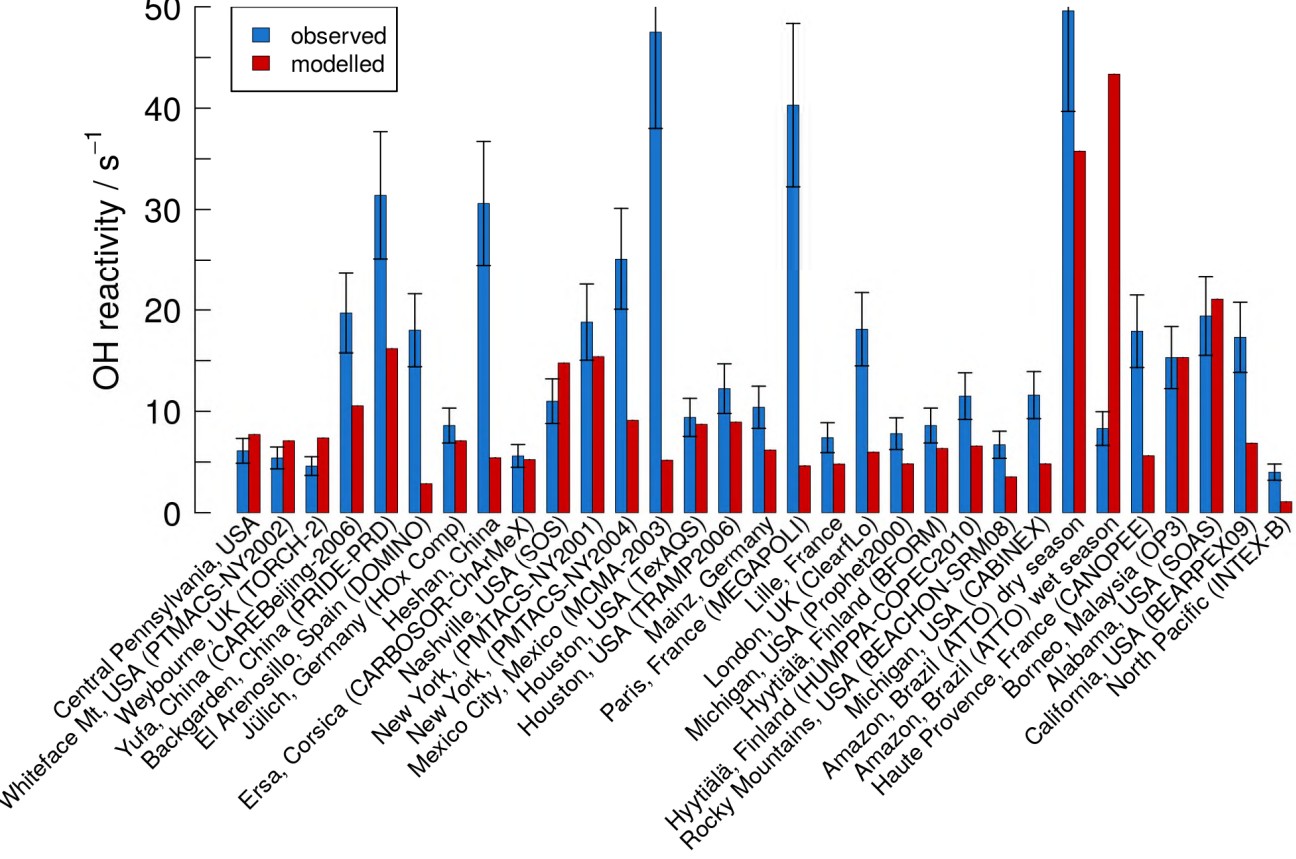

**Figure 2: Comparison of the average observed $k_{OH}$ with modelled $k_{OH}$. Error bars (20%, relative) are commensurate with the measurement uncertainty of the observed $k_{OH}$.**




Observed $k_{OH}$ is compared to that simulated by UM-UKCA for the same longitude, latitude and month in Figure 2. Observed and modelled $k_{OH}$ agree within 20 % of one another in 12 out of 27 cases. Of the remaining 15 campaigns, the model underestimates $k_{OH}$ in 14 cases and overestimates it in only one. That the model underestimates the total $k_{OH}$ is not a surprise; if the missing sink was an overlooked primary emission, it would not be accounted for in the model, whereas if the missing

5   reactivity arose from oxidation intermediates, these would also be underestimated by the model, as the oxidation schemes of large VOCs (*e.g.*, isoprene) are somewhat simplified in the CheST scheme compared to a complete mechanism, and also because only a limited number of VOCs are emitted in the model compared to the atmosphere. In the one case in which the model significantly overestimated $k_{OH}$ compared to the measurements (Amazon ATTO, during the wet season), this was mainly the result of high levels of isoprene in the model (see below).

10   Even in the cases in which the model and observations are in good agreement, it is important to ascertain that the modelled OH reactivity is indeed the result of the same sinks observed in the field; for this purpose, individual reactivities measured in each campaign provide a large amount of information that can be used to establish how well the model captures the speciation of $k_{OH}$. This is shown in Figure 3, where the modelled reactivities from the known OH sinks are plotted against those measured in the field.

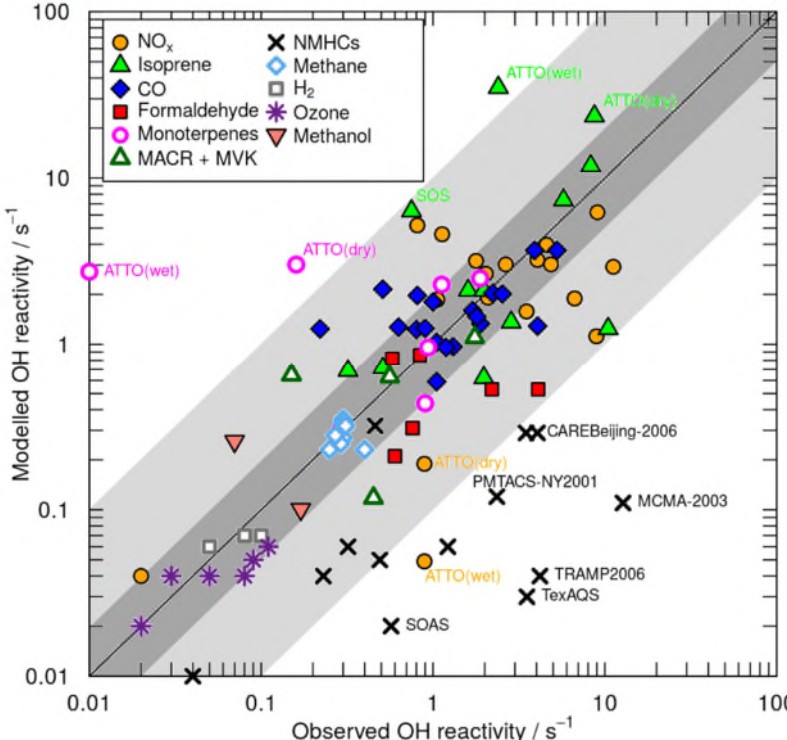

**Figure 3: Scatter plot of modelled OH reactivity arising from known OH sinks against measurements. Also shown is the 1:1 line (solid black line) as well as the factor-of-2 and factor-of-10 deviations from it (dark grey and light grey areas respectively). Specific data points that are discussed in the text are labelled with the name of the field campaign in which those particular measurements were performed.**





We see overall a reasonably good agreement between modelled and observed sinks, albeit with some scatter within an order of magnitude. The main source of discrepancy between modelled and observed individual reactivities are differences in the number densities of the OH sinks ([$X_i$] in Eq.(1)) and not the temperature-dependent rate constants, as temperature differences

between the model and the measurements only have a minor effect on the rate constants used to calculate the reactivities. The abundance of each individual OH sink species is determined by the balance between its sources (*e.g.*, emissions) and sinks (largely, reaction with OH). It is however difficult to establish whether the differences between observed and modelled OH sinks arise from misrepresenting emissions or abundances of the hydroxyl radical itself without comparing modelled and observed [OH], and measurements of the OH concentrations are only available for a small subset of the campaigns considered

here.

Figure 3 provides a useful guide on the magnitude of the contribution of different OH sinks to the total $k_{OH}$. The main contributors to $k_{OH}$, with reactivities ranging between 1 and 10 s$^{-1}$, are isoprene in forested environments and NO$_x$, CO, formaldehyde and non-methane hydrocarbons (NMHCs, indicating primarily alkanes and alkenes) in urban environments. At

the other end of the spectrum, species such as methane, ozone and hydrogen only give rise to small contributions (< 1 s$^{-1}$) to the total $k_{OH}$.

The speciation of the total OH reactivity shown in Figure 3 allows us to investigate the reasons for the discrepancy between some of the modelled $k_{OH}$ and observations highlighted in Figure 2. For instance, the disagreement between modelled and

observed $k_{OH}$ in some urban environments (notably Mexico City, wintertime New York, Houston and Beijing/Yufa) is almost entirely attributable to the underrepresentation of NMHCs in the model. This can be accounted for in terms of species lumping. As with many state-of-the-art models, instead of adding numerous hydrocarbons to the emission and chemistry schemes, the heavier alkanes and alkenes were lumped into the emission fields of ethane and propane, weighted by carbon number. We can see that lumping serves as a reasonable approximation for the representation of the abundance of some carbon-containing

species (such as CO and formaldehyde, the ultimate products of hydrocarbon oxidation, which are in reasonable agreement with observations as shown in Figure 3 and also in Figure S1 in the Supplementary Material for CO). However lumping introduces an additional complication when the OH reactivity of NMHCs is calculated. As the reactivity is defined as the product of the rate constant for the reaction of the NMHCs with OH and the number density of the NMHCs, and as the rate constants for the reaction of OH with ethane and propane are 1-2 orders of magnitude smaller than those of OH with higher

alkanes (C $\geq$ 4) and alkenes (C $\geq$ 2), lumping leads to an underestimate of the same magnitude in the reactivity of the NMHCs.

Figure 3 also offers an explanation for the only instance in which the model significantly over predicted $k_{OH}$. The abundance of isoprene measured during the wet season of the ATTO campaign in the Amazon (~1 ppbv, or nmol/mol, in March 2013) was more than an order of magnitude lower than that predicted by the model for the same time of the year (~14.6 ppbv). As



discussed above, this might arise from either overestimated isoprene emissions or from underestimated OH abundances in the model. As OH concentrations were not measured during the ATTO campaign, a direct comparison of modelled and observed [OH] is not possible. However the abundance of other short-lived OH sinks is also overestimated by the model; notably, the observed concentration of monoterpenes (reported to be below the detection limit of the PTR-MS used by Nölscher and co-

workers, and here approximated to 0.01 ppbv) was much lower than in the model (2.2 ppbv). This might be the result of the model underestimating the secondary OH originating from the oxidation of large organics (*e.g.*, isoprene and monoterpenes, as described in Archibald et al., 2010). In this specific instance the model also underestimated the concentration of NO (34 pptv, or pmol/mol, *vs* the observed ~1 ppbv), which might have limited the production of secondary OH via the reaction of $HO_2$ with NO relative to observations.

The methane lifetime with respect to tropospheric loss *via* reaction with OH, $\tau_{CH_4}$, for the base run was 8.75 years, which is within $1\sigma$ of the ACCMIP multimodel mean for the year 2000 (9.7 ± 1.5 years). $\tau_{CH_4}$ from the base run is also in good agreement with the value of 8.5 years reported by Lelieveld et al. (2016). Notably the model used by these authors exhibited some differences from the one used in the current work: Lelieveld et al. (2016) used emissions for the year 2010, defined the

tropopause *via* their own diagnostic and employed an extensive chemistry scheme consisting of 1630 reactions. Notwithstanding these differences, the values for $\tau_{CH_4}$ from the two studies are in very good agreement. In the Base run, the average lifetime of the OH radical, $\tau_{OH}$, was 1.18 s for the whole troposphere, 0.57 s within the boundary layer and 0.45 s at the surface, as summarised in Table 4.

## 4 Modelling the missing reactivity: addition of sink X

Figure 3 highlights that whilst the model represents many of the individual components of OH reactivity within at least an order of magnitude (and often within a factor of two) of observations, the model underrepresents total $k_{OH}$ in the majority of the cases (Figure 2). There are a number of ways to account for this missing reactivity in the model. For example, additional species (such as more reactive NMHCs or more reactive reaction intermediates) and their chemistry could be included in the model. However observations indicate that this approach would still leave outstanding missing reactivity (Edwards et al., 2013;

Elshorbany et al., 2012; Kaiser et al., 2016; Kovacs et al., 2003; Lee et al., 2009; Lou et al., 2010; Mao et al., 2012; Mogensen et al., 2011; Yang et al., 2017). In this work, we have taken a different, simpler approach. A new species representing a direct sink of OH was added into the model and its atmospheric implications were assessed. Emissions of the unspecified OH sink, species X, were introduced in the model simulations along with the reaction:

X + OH           →       products,                 (R3)





with a temperature independent bimolecular rate constant, $k_3 = 1 \times 10^{-10}$ cm$^3$ molecules$^{-1}$ s$^{-1}$, set to represent reactions with a very reactive compound (*i.e.*, OH + reactive VOCs). Crucially, the global and seasonal abundance of X was underpinned by field observations of missing $k_{OH}$. This section discusses the implementation of this scheme in the model and its effects.

## 4.1 Generating a global field of missing reactivity

To generate a surface field for the missing reactivity, multiple linear regression was applied. This method consisted of fitting the missing reactivity from observations to the corresponding model grid box emissions of VOCs, NO$_x$ and aerosol precursors at the individual observation sites where the OH reactivity was measured (described in Section 2). This resulted in a time-varying spatial field for the missing reactivity based on the predictors (emission fields) listed in Table 3. Correlation of the missing $k_{OH}$ with some of the emitted species would be expected both if the missing sink was an oxidation intermediate (in which case it would correlate with its precursors) and if it was a primary emission (in which case it could be expected to correlate to other primary emitted VOCs such as anthropogenic or biogenic VOCs). This is also supported by observations. For instance, measurements taken during the INTEX-B campaign (Mao et al., 2009) found that the missing reactivity correlated with formaldehyde concentrations; the authors concluded that this indicated that the missing reactivity potentially arose from VOCs that formed formaldehyde upon oxidation. Similarly, measurements performed in a forest in Michigan during the CABINEX campaign showed good correlation between the missing reactivity and both isoprene and its oxidation products (methyl vinyl ketone, MVK, and methacrolein, MACR) when the missing reactivity was highest (Hansen et al., 2014).

The results of the multiple linear regression using all 15 predictors in Table 3 are shown in Figure 4; the $R^2$ value for the fit was 0.75. The strongest predictors for the missing reactivity were the emissions of organic carbon from biofuels (OC biofuel), black carbon from biofuels (BC biofuel), acetone and CO. Whilst these might indicate both an anthropogenic and a biogenic component of the missing reactivity, none of the predictors had a $p$-value $\leq 0.05$ (with OC biofuel coming close with $p = 0.067$), perhaps as a result of the small sample size available. The coefficients from the multiple linear regression are shown in Table 3. The fact that the strongest predictors were the emissions for aerosol tracers not included in the model gaseous chemistry might also indicate potential contributions to the total $k_{OH}$ from condensed-phase particles. However the role of particulate matter in OH loss is very poorly characterised, as highlighted in the recent review by Yang et al. (2016).

To establish the robustness of the outcome of the multiple linear regression routine, the analysis was repeated using different subsets of predictors. Unsurprisingly, the iterations using larger numbers of predictors returned better error statistics ($R^2$ values, normalised mean biases, *etc.*). The inclusion of the bottom three predictors in Table 3 (biogenic methanol, ethane and propane emissions) led to only marginal improvements in the quality of the fit (*e.g.*, increases in $R^2 < 0.1\%$) in all cases. On the other hand the top four entries in Table 3 were the strongest predictors in all iterations that included them and their exclusion from the fitting routine affected the quality of the fit significantly (*e.g.*, decreases in $R^2 > 10\%$) . For the purpose of this work, the





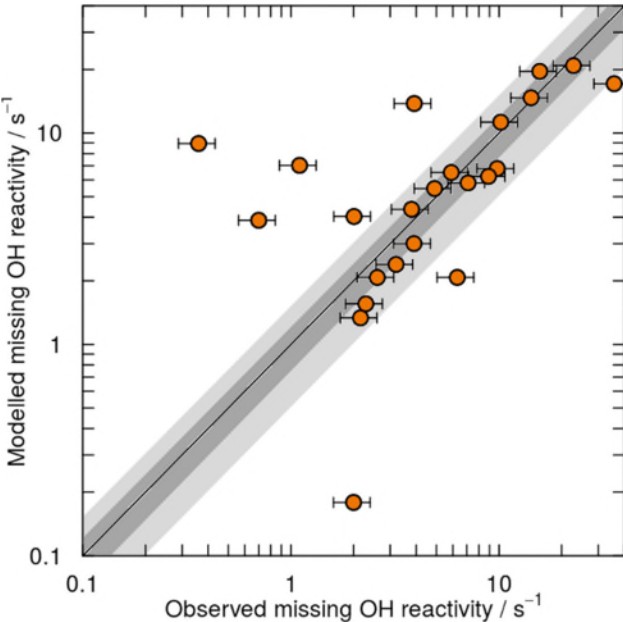

**Figure 4: Scatter plot of missing OH reactivity modelled by multiple linear regression against observed missing OH reactivity (as reported in Table 2). Each point represents one observation site. The error bars (20%, relative) are commensurate to the uncertainty in the $k_{OH}$ measurements in the field (Yang et al., 2016). Also shown are the 1:1 line (solid line), as well as 20% and 50% deviations**
5  **from it (dark grey and light grey areas respectively). The $R^2$ value from the multiple linear regression was 0.75.**

outcome of the multiple linear regression using all 15 predictors was used.

The multiple linear regression resulted in an expression of missing reactivity at the surface varying with longitude, latitude
10  and time. Figure 5 shows global seasonal averages of the modelled missing reactivity at the surface in the boreal winter (DJF)
and summer (JJA). The multiple linear regression captured some of the seasonality of the missing reactivity over forested
regions at mid-latitudes: this behaviour is consistent with the temperature dependence of the missing reactivity observed in a
number of field campaigns in forested areas (as described in Section 1). In addition, a number of urban areas are also
distinguishable as regions of high missing reactivity.





**Table 3: Outcome of the multiple linear regression (MLR); the predictors are sorted by increasing *p*-value.**

| Predictor | Coefficient/$10^{12}$ kg$^{-1}$ m$^2$ |
|---|---|
| OC biofuel emissions | 4.41 |
| BC biofuel emissions | -12.97 |
| Acetone emissions | 0.78 |
| CO emissions | -0.02 |
| OC fossil fuel emissions | -0.67 |
| BC fossil fuel emissions | 1.25 |
| Monoterpene emissions | 0.04 |
| NO$_x$ emissions | -0.02 |
| Isoprene emissions | -0.02 |
| Methane emissions | 0.01 |
| Acetaldehyde emissions | -1.18 |
| Formaldehyde emissions | 1.20 |
| Biogenic methanol emissions | -0.02 |
| Ethane emissions | -0.22 |
| Propane emissions | 0.76 |





**Figure 5: Global distribution of the simulated missing OH reactivity (in s⁻¹) in the boreal winter (DJF, top panel) and boreal summer (JJA, bottom panel).**

5   In order to establish the effects of the additional sink on tropospheric oxidation chemistry it is necessary to emit species X in the model and let it interact with OH *via* R3. However the conversion of the global missing reactivity derived from the multiple linear regression into an emission field is not straightforward. Initially, the emission rate of X is calculated as equal to the rate of removal of X within the turbulent boundary layer for each surface grid cell, according to Eq. (2):





$$emission\ rate\ of\ X\ /\ \mathrm{kg\ m^{-2}s^{-1}} = k_3[\mathrm{X}][\mathrm{OH}] \times \frac{h\,M_r}{N_A} \times 10^3 \qquad (2)$$

where $k_3$ is the rate constant for R3 (in $\mathrm{cm^3\ molecules^{-1}\ s^{-1}}$), [X] and [OH] are the number densities of X and OH respectively

(in units of molecules $\mathrm{cm^{-3}}$), $h$ is the height of the turbulent boundary layer (in m), $M_r$ is the molar mass of species X (arbitrarily

assigned a value of 30 g $\mathrm{mol^{-1}}$) and $N_A$ is Avogadro's number (in molecules $\mathrm{mol^{-1}}$). The term $k_3[\mathrm{X}]$ corresponds to the missing

reactivity from the multiple linear regression, while the values of [OH] are taken from the base run. The model was run for

one year with these emissions of X. Then the OH reactivity arising from the newly emitted species X was calculated for the

model run (as $k_3[\mathrm{X}]$) and compared with the missing reactivity obtained from the multiple linear regression. It was found that

the OH reactivity arising from species X in this initial model run was significantly higher than the missing reactivity determined

*via* multiple linear regression (as shown in Figure 6b). A routine was therefore developed, in which the initial emission field

of species X was iteratively optimised in a series of model runs until the OH reactivity from species X in the model matched

the missing reactivity determined *via* multiple linear regression within the tolerances specified in Figure 6a. The procedure

used is summarised in Figure 6. As the missing reactivity from the multiple linear regression was underpinned by observations,

this routine ensured that the additional sink emitted in the model would still account for the observed missing reactivity.





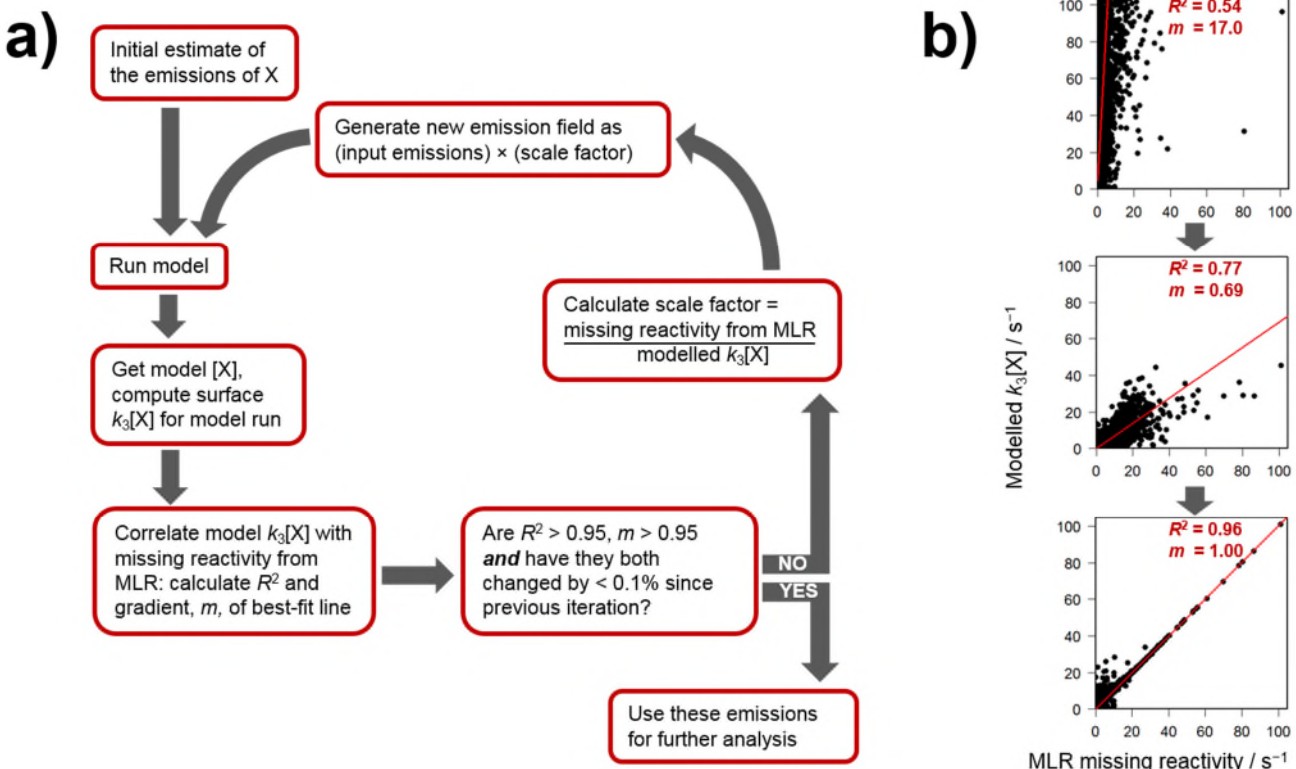

**Figure 6: Implementation of the routine used to convert the global missing reactivity field obtained from multiple linear regression (MLR) into emissions of species X. Panel a) illustrates the procedure as a flow chart, while panel b) shows the correlation plots between modelled $k_3[X]$ and the MLR missing reactivity from non-consecutive iterations: as the $R^2$ values and the gradients of the correlation plots converge to 1, the emissions of X in the model lead to X-reactivities identical to the missing reactivity derived by MLR.**

### 4.2 Effects of introducing sink X

The chemistry used for sink X does not allow any secondary OH formation following the initial oxidation of X in R3. This is unrealistic, as the oxidation of the vast majority of trace species leads to some degree of secondary OH production; however, with so little information available on the identity of species X, the system was too poorly constrained to even attempt an educated guess on the OH recycling probability of the products of R3. Therefore the effects of introducing sink X in the model, discussed at length in this Section, can be seen as a 'worst case' scenario, one in which no OH is regenerated following R3 and in which OH is removed from regions of high emissions of X. This may be the case for reactions of OH on aerosol surfaces.

OH abundances are reduced following inclusion of R3 in the chemistry scheme, as a result of both direct removal of OH *via* R3 and less efficient production of secondary OH (as highlighted in the fluxes reported in the Supplementary Material). As shown in Figure 7, the most OH-depleted areas at the surface include Scandinavia, Eastern Europe and the coastlines of the





Persian Gulf, Venezuela and Java. Peak OH reductions in these regions are of the order of 5-6 $\times$ $10^6$ molecules cm$^{-3}$. In these regions as much as 90% of the mean annual [OH] is removed, but it must be stressed that secondary OH production from the products of R3 would mitigate these effects. Interestingly some of the regions affected by the highest emissions of X (*i.e.*, the Amazon region and Central Africa) only exhibit relatively small decreases in OH concentrations, down by ~20 %; as these

5   areas are rich in OH sinks, the introduction of an additional sink does not affect the OH budget significantly. OH depletion is most pronounced at the surface and in the boundary layer in the Northern Hemisphere (also in Figure 7), which is consistent with the vertical distribution of the additional OH reactivity brought about by species X shown in Figure 8. Mean tropospheric OH decreased by 1.6 %, while mean OH abundances in the boundary layer and at the surface were reduced by 5.6 and 8.1 % respectively. Seasonal changes in OH abundances, both in absolute and relative terms, are shown in Figures S2 and S3 in the

10   Supplementary Material. These show significant relative reductions in the mean boreal winter OH in the Northern

**Figure 7: Annual mean change in OH concentration following inclusion of R3: absolute change in $10^6$ molecules cm$^{-3}$ a) at the surface and b) as a zonal mean, and relative percentage change c) at the surface and d) as a zonal mean.**





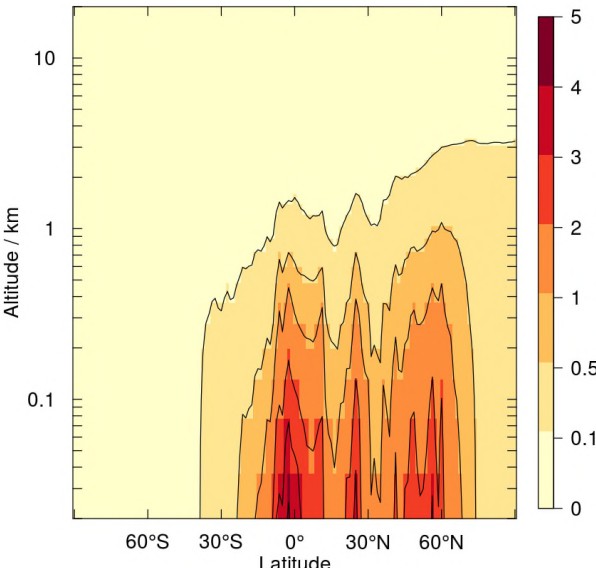

**Figure 8: Annual zonal mean of the OH reactivity (in s$^{-1}$) arising from sink X.**

Hemisphere ($> 60\,\%$), which are however very small in absolute terms ($< 5 \times 10^4$ molecules cm$^{-3}$).

5   $\tau_{CH_4}$ for the model run including R3 was 8.95 years (~2.3 % higher than the base run). $\tau_{OH}$ was reduced by approximately 2 %
at the surface, by 3% in the boundary layer and by 1.5 % in the whole troposphere (Table 4). Another metric of interest when
discussing tropospheric oxidation chemistry is the OH recycling probability, $r$, which describes the sensitivity of the OH
chemistry to perturbations (Lelieveld et al., 2002). $r$ was calculated in accordance with Lelieveld et al. (2002, 2016), using Eq.
(3):

$$r = 1 - P/G \qquad\qquad\qquad (3)$$

where $P$ is the rate of formation of primary OH (*i.e.*, *via* R1 and R2) and $G$ is the gross OH formation rate, consisting of the
sum of $P$ and the formation rate of secondary OH, $S$ (*i.e.*, *via* any route other than R1 and R2). Using the OH formation fluxes
15   tabulated in the Supplementary Material, it was found that $r$ remained larger than 60 % in both runs, indicating that tropospheric
oxidation was effectively 'buffered'.





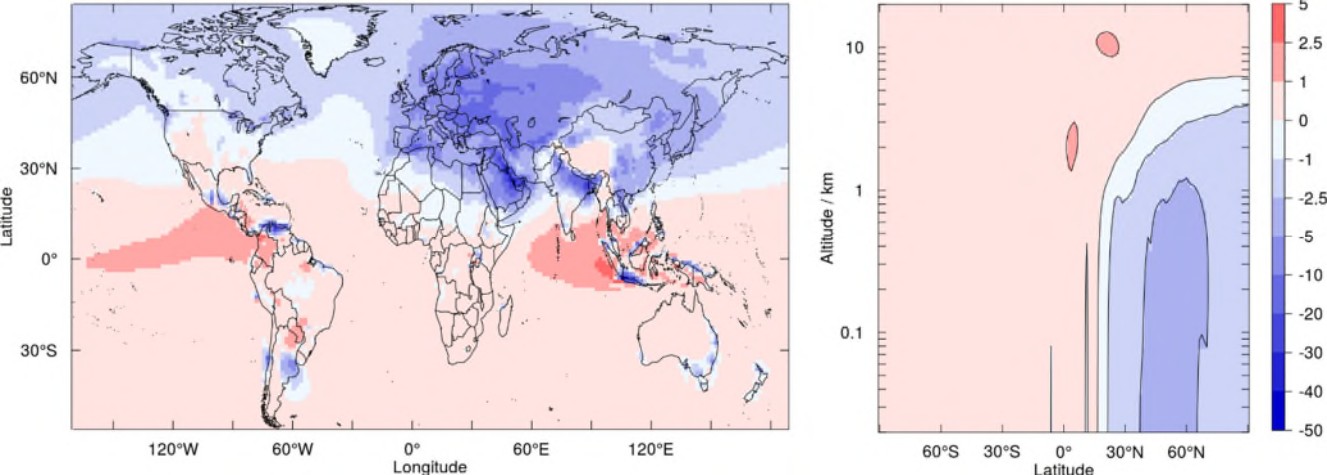

**Figure 9: Annual mean percentage change in O₃ concentration at the surface (left) and as a zonal mean (right).**

The ozone burden at the surface decreased slightly by 0.9%, but the overall tropospheric ozone burden increased by 0.3 %. This somewhat contradictory behaviour in ozone, also shown in Figure 9, can be accounted for in terms of the changes brought

about by decreases in OH in regions with different $NO_x$ abundances. In general, OH decreases are accompanied by lower concentrations of $HO_2$ and peroxy radicals ($RO_2$); in OH-depleted, $NO_x$-rich regions (*e.g.*, Europe in Figure 9) less $HO_2$ and $RO_2$ are available to react with NO, leading to a reduction in ozone produced from $NO_2$ photolysis. On the other hand, in remote regions with low $NO_x$ the lower abundance of $HO_2$ and $RO_2$ following sequestration of OH by X leads to less efficient ozone removal *via* the reaction of $O_3$ with $HO_2$ (and, to a lesser extent, *via* the reaction of $HO_2$ with $RO_2$, producing soluble

alkyl hydroperoxides), which appears as a slight positive change in Figure 9. Decreases in surface ozone in the Northern Hemisphere are also exacerbated by the absence of further chemistry following R3. This rules out the production of X-peroxy radicals from the initial oxidation of X, which in turn would react with NO to produce $NO_2$, the photolysis of which would then lead to $O_3$ formation. However to what degree the ozone changes shown in Figure 9 would be mitigated by subsequent chemistry would be highly dependent on the actual nature of the missing sink. Comparison of ozone seasonal observations

from a number of remote sites with the model output (illustrated in Figure S4 in the Supplementary Material) shows that, while generally the model and the observation are in good agreement in a number of the locations shown, the inclusion of the additional OH sink X has a minimal impact on the modelled ozone at these locations.

On average, R3 accounts for approximately 6 % of the total OH loss flux at the surface (see Supplementary Material); in some

particular regions, such as the Amazon rainforest, this contribution increases to up to 20 %, whilst in areas of Eastern Europe it went up to 50 % (see Figure S5 in the Supplementary Material). As a result, reactions of OH with other sinks (*e.g.*, methane) become less efficient and the lifetime of such species increases: the lifetime of isoprene, a major tropospheric OH sink, increases by 17% from 2.36 hours to 2.76 hours. This is reflected in the reduced flux through its reaction with OH in the



regions of high X reactivity (Tables S1-S3). As a result of its longer lifetime, isoprene is transported to higher altitudes than in the base run, as the flux through its reaction with OH increases above the boundary layer.

**Table 4: Values of the OH and methane lifetimes in the model runs described in this work. Numbers in brackets indicate the percentage change with respect to the base run.**

|  | Base run | X + OH run | CH₃O₂ + OH run 1 | CH₃O₂ + OH run 2 | CH₃O₂ + OH run 3 |
|---|---|---|---|---|---|
| $\tau_{OH}$ surface / s | 0.455 | 0.447 (−1.8%) | 0.442 (−2.7%) | 0.442 (−2.7%) | 0.442 (−2.7%) |
| $\tau_{OH}$ boundary layer / s | 0.574 | 0.557 (−3.2%) | 0.558 (−2.8%) | 0.558 (−2.8%) | 0.558 (−2.8%) |
| $\tau_{OH}$ troposphere / s | 1.18 | 1.16 (−1.5%) | 1.15 (−2.5%) | 1.15 (−2.5%) | 1.15 (−2.5%) |
| $\tau_{CH_4}$ / years | 8.75 | 8.95 (+2.3%) | 8.92 (+2.0%) | 8.97 (+2.5%) | 9.01 (+3.0%) |

While the work described in this Section can be considered a worst-case scenario due to the absence of OH recycling following R3, it is important to observe that the impacts on the wider atmosphere are relatively minor. Reconciling the observations of missing OH reactivity in our model in a worst-case scenario (for OH) has little impact on the global oxidising capacity. We report a small increase in methane lifetime, and an overall minor decrease in surface ozone; however the actual extent of these changes in the real atmosphere is likely to be mitigated to some degree by OH recycling following the initial oxidation of the missing sink.

## 5 Effects of RO₂ + OH chemistry

The previous Section focused on simulating the observed missing reactivity by introducing an additional sink in the form of species X. Based on structural-reactivity arguments, Archibald et al., (2009) postulated that there could be a reaction between peroxy radicals (RO₂) and OH, which could act as a sink for OH and, depending on the mechanism, a potential source of oxygenated VOC. A lack of any experimental data hampered estimations of the rate constant for the reaction. However recent laboratory studies have confirmed that peroxy radicals do indeed react with OH and, subsequently, the kinetics of the simplest RO₂ + OH reactions have been characterised (Fittschen et al., 2014). The potential impact of RO₂ species as OH sinks and their contribution to the total OH reactivity have not been investigated to date.

### 5.1 Overall contribution of modelled RO₂ to $k_{OH}$

The UM-UKCA base model described in Section 2 includes the formation and reactions of a number of peroxy radicals. This includes not only those originating from the oxidation of the simplest hydrocarbons (alkanes and carbonyl-containing species up to three carbon atoms), but also first and second generation peroxy radicals produced in the oxidation of isoprene. This allowed the calculation of an additional term for $k_{OH}$ arising from the contributions of all RO₂ radicals in the model, $k'_{OH}$, as shown in Eq. (4):



$$k'_{OH} = \sum_{i=1}^{n} k_{OH+RO_{2,i}} [RO_{2,i}], \qquad\qquad (4)$$

where $[RO_{2,i}]$ is the concentration of peroxy radical $i$ and $k_{OH+RO_{2,i}}$ is the rate constant of its reaction with OH. Assumptions had to be made on the individual values of $k_{OH+RO_{2,i}}$, as only the rate constants of the reactions of OH with methyl (Assaf et al., 2016, 2017b; Bossolasco et al., 2014; Yan et al., 2016), ethyl (Assaf et al., 2017a; Faragó et al., 2015), propyl (Assaf et al., 2017a) and $i$- and $n$-butyl (Assaf et al., 2017a) peroxy radicals have been measured in the laboratory to date. Reported experimental values for the rate constant of the reaction of the simplest peroxy radical, $CH_3O_2$, with OH disagree by more than a factor of 3. The highest value to date, reported by Bossolasco et al. (2014) ($2.8 \times 10^{-10}$ cm$^3$ molecule$^{-1}$ s$^{-1}$ at $T = 294$ K), has since been revised to a lower value ($1.6 \times 10^{-10}$ cm$^3$ molecule$^{-1}$ s$^{-1}$ at $T = 295$ K) by the same group (Assaf et al., 2016): the difference is thought to arise from complicating secondary chemistry due to the presence of electronically excited iodine atoms following the photolysis of the gaseous mixture used by Bossolasco et al. (2014). However an even lower value has been reported by Yan and co-workers ($8.4 \times 10^{-11}$ cm$^3$ molecule$^{-1}$ s$^{-1}$ at $T = 298$ K), and the reason for the discrepancy between this study and that by Assaf et al. is still unclear.

In general, the rate constant of these reactions exhibit no significant dependence on the size of the alkyl group on the $RO_2$ radical. For the purpose of this study all $k_{OH+RO_{2,i}}$ were assumed to be equal to the rate constant of the reaction of the methyl peroxy radical with OH at $T = 295$ K as measured by Assaf et al. (2016), and independent of temperature. Eq. (4) therefore can be re-written as:

$$k'_{OH} = k_{OH+RO_2} \sum_{i=1}^{n} [RO_{2,i}], \qquad\qquad (5)$$

where $k_{OH+RO_2} = 1.6 \times 10^{-10}$ cm$^3$ molecule$^{-1}$ s$^{-1}$.

The additional annual mean OH reactivity at the surface resulting from $RO_2$ chemistry is shown in Figure 10: while the largest contributions in absolute terms to the total $k_{OH}$ are found in forested tropical regions (Figure 10a, where $RO_2$ radicals from isoprene dominate), these are indeed very small when compared to the total reactivity present in the same regions. In relative terms, $RO_2$ radicals are most significant as OH sinks over remote tropical oceans (Figure 10c), where the majority ($> 90\%$) of the $RO_2$ contribution to $k_{OH}$ arises from the methyl peroxy radical. This contribution extends beyond the boundary layer and into the free troposphere over tropical latitudes, as shown in Figure 10b and Figure 10d.





**Figure 10: changes in total $k_{OH}$ from the inclusion of RO$_2$ + OH reactions: a) global surface change in s$^{-1}$ and b) zonal mean in s$^{-1}$; c) percentage change at the surface and d) percentage change zonal mean.**

## 5.2 The CH$_3$O$_2$ + OH reaction and product branching simulations

Recent studies on the products of the reaction of the simplest peroxy radical, CH$_3$O$_2$, with OH may help establish the wider atmospheric implications of this novel peroxy radical chemistry. Three product channels can be envisaged for this reaction (Archibald et al., 2009):

CH$_3$O$_2$ + OH $\rightarrow$ HO$_2$ + CH$_3$O,       (R4a)

$\rightarrow$ H$_2$O + CH$_2$O$_2$,       (R4b)

$\rightarrow$ O$_2$ + CH$_3$OH,       (R4c)





with branching ratios defined as:

$$\alpha = k_{4a}/(k_{4a} + k_{4b} + k_{4c}), \tag{5a}$$

$$\beta = k_{4b}/(k_{4a} + k_{4b} + k_{4c}), \tag{5b}$$

$$\gamma = k_{4c}/(k_{4a} + k_{4b} + k_{4c}). \tag{5c}$$

Recent laboratory studies identified R4a as the major product channel, with $\alpha = 0.80 \pm 0.20$ (Assaf et al., 2017b). Channel R4b, producing the Criegee intermediate $CH_2O_2$, was found to be a minor contributor to the overall reaction ($\beta < 0.05$). As the set-up used in the study was not suitable for the detection of methanol ($CH_3OH$), the magnitude of $\gamma$ could not be established. A theoretical study identified R4c as a potentially significant source of methanol in the remote boundary layer and modelled its impacts (Müller et al., 2016). However this study predated the first (and, so far, only) experimental determination of the products of R4, and also used the very high value of $k_4$ reported by Bossolasco et al. (2014), which has since been revised to a value almost a factor of 2 lower than the original (as discussed in Section 5.1).

In the current work, three simulations were run with different sets of values for $\alpha$ and $\gamma$ (run 1: $\alpha = 1$, $\gamma = 0$; run 2: $\alpha = 0.8$, $\gamma = 0.2$; run 3: $\alpha = 0.6$, $\gamma = 0.4$) in order to establish the atmospheric implications of different product branching for R4 over the uncertainty range of $\alpha$ reported by Assaf et al (2017b).

As shown in Table 4, introduction of R4 in the model led to shorter $\tau_{OH}$ (by approximately 3 %), regardless of the product branching. Tropospheric methane lifetime increased by as much as 3 % in run 3. $HO_2$ abundances increased in all runs and in particular in run 1, which exhibited $HO_2$ concentrations higher than the base run by as much as 12 % over remote oceans, as shown in Figure 11. Mean tropospheric $HO_2$ abundances increased by 3.9 % in run 1, 2.8 % in run 2 and 1.7 % in run 3.

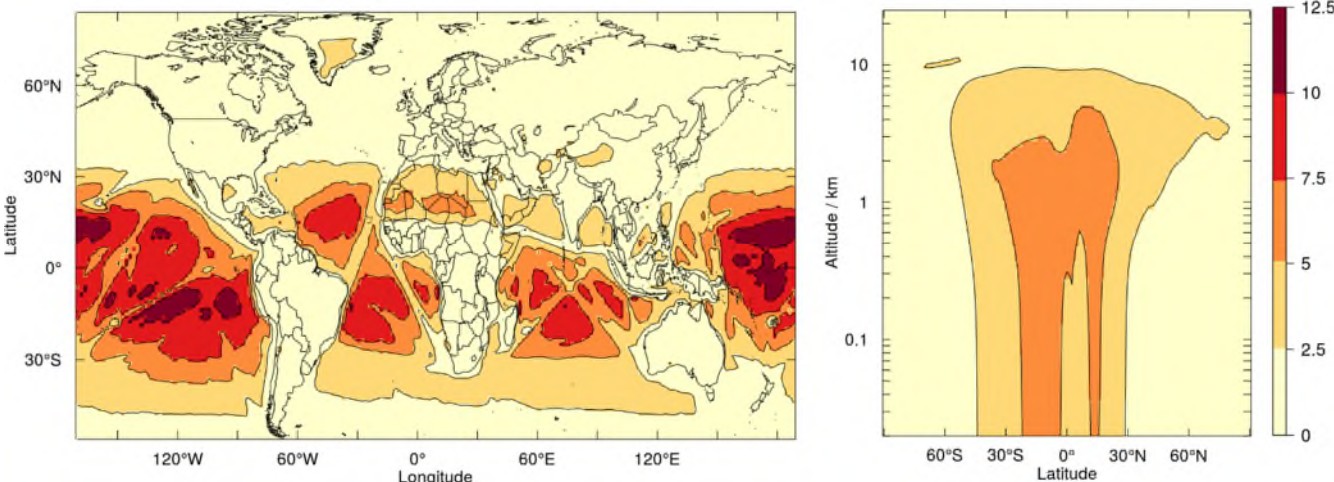

Figure 11: Annual mean percentage change in $HO_2$ concentrations at the surface (left) and as a zonal mean (right).



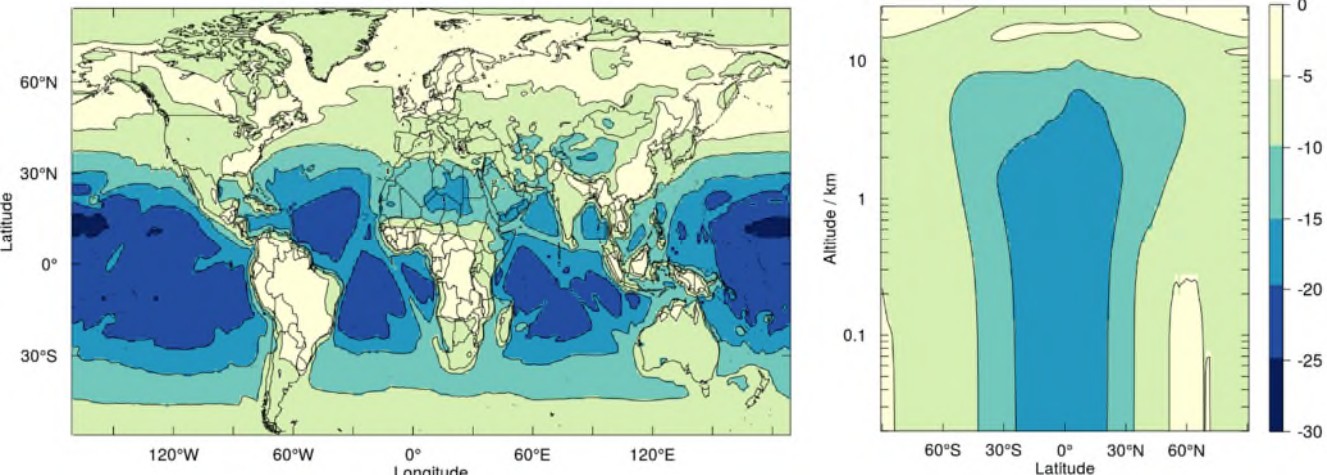

**Figure 12: Annual mean percentage change in CH$_3$O$_2$ concentrations at the surface (left) and as a zonal mean (right).**

Regardless of the product branching of R4, the concentration of methyl peroxy radicals at the surface decreased significantly (by as much as 30%) over remote oceans and more moderately (by 5-10%) over land at mid-latitudes (Figure 12). Mean tropospheric CH$_3$O$_2$ abundances decreased by 14 % in all runs, whilst mean tropospheric OH was reduced by 1.5 % in run 1, 2.1 % in run 2 and 2.7 % in run 3.

The inclusion of R4 in the model led to a small reduction ($\sim -1$ %) in the tropospheric ozone burden. This is mainly driven by the increase in HO$_2$ abundances, leading to enhanced ozone removal *via* O$_3$ + HO$_2$ over remote oceans. Ozone abundances over NO$_x$-rich areas were largely unchanged, as the reaction of CH$_3$O$_2$ with NO dominates over R4 and therefore HO$_2$ concentrations did not deviate significantly from the base run.

The largest difference in the impact of different branching ratios for R4 was observed for methanol concentrations. In the scenario in which $\alpha = 1$ (and $\gamma = 0$), methanol concentrations decrease by as much as 40 % over remote oceans as R4a efficiently inhibits methanol production *via* the CH$_3$O$_2$ self-reaction. However increasing $\gamma$ increases methanol mixing ratios. When $\gamma =$ 0.4 methanol abundances are enhanced by up to 200 % (relative to the base case) over remote regions. Mean tropospheric methanol decreased by 8.4 % in run 1 and increased by 35.9 and 80.2 % in runs 2 and 3 respectively. There remains uncertainty in the tropospheric methanol budget (Khan et al., 2014; Millet et al., 2008), with models currently underestimating atmospheric methanol concentrations significantly. Müller and co-workers suggested that a scenario with high methanol yield from R4 could reconcile models with observation, based on a model run with effectively $\gamma = 0.4$ which produced an additional 117 Tg year$^{-1}$ of methanol (Müller et al., 2016). However these simulations used the high value of $k_4$ reported by Bossolasco et al. (2014). If the preferred lower value reported more recently by Assaf et al. (2016) is used with $\gamma = 0.4$, we calculate methanol production *via* R4c to be 60 Tg year$^{-1}$ and we estimate that a value of $\gamma$ greater than 0.8 is needed to produce the amount of methanol necessary to reconcile models with observations. This is well beyond the uncertainty in the laboratory measurements



of the branching of R4 and highlights that missing sources of methanol must come from direct emissions (or re-emissions) or as yet-undiscovered photochemical sources, rather than from the reaction between OH and $CH_3O_2$.

Results from this work agree with the box model calculations performed by Assaf et al. (2017b), indicating that the largest impacts of R4 are on $HO_2$ and $CH_3O_2$ abundances. In addition, we show that these effects are not limited to the surface or the boundary layer but also extend well into the free troposphere.

## 6 Conclusions

The hydroxyl radical plays a pivotal role in the chemistry of the atmosphere. Its abundance determines the lifetime of most emitted compounds so that OH is often known as the "atmospheric detergent". However, our understanding of the chemistry and distribution of OH is far from complete. This study has examined the total tropospheric OH reactivity, $k_{OH}$, using the UM-UKCA chemistry-climate model. In the first instance, the model was evaluated against available measurements of known OH sinks. This comparison indicated that, while the model captured the abundances of a number of known OH sinks reasonably well in a variety of regions across the planet, the total modelled OH reactivity was generally much lower than observed, and there are significant biases in the model's ability to accurately simulate reactivity from NMHCs. Partly, this error was linked to the limited NMHC chemistry included in Chemistry-Climate models like UM-UKCA.

Existing observations of the missing OH reactivity were used to develop a method to account for the missing OH sink in the model by introducing an additional reaction and OH sink species, X, in the model chemistry scheme. Observations of missing reactivity were correlated with underlying inputs into the model (emissions of VOCs, $NO_x$ and aerosol precursors) through multiple linear regression analysis. The multiple linear regression fit highlighted correlation with both biogenic and anthropogenic emissions, consistent with observations of missing reactivity in remote and urban environments. The fitting routine also indicated strong correlation of the missing reactivity with the emissions of particulate matter, perhaps pointing at OH loss processes involving condensed-phase particles that have been overlooked to date. Our simulations showed that the largest impacts of the global missing OH reactivity were at the surface and in the boundary layer, where sink X accounted on average for 6 % of the total OH loss flux. Inclusion of X in the model led to decreases in mean OH abundances of 8.1% at the surface, 5.6 % in the boundary layer and 1.6 % in the whole troposphere. Inclusion of missing reactivity, in the form of X, reduces and increases the lifetimes of OH and methane, respectively, by approximately 2 %. The inclusion of X modifies the global ozone burden only slightly (< 1 %) but has larger impacts on simulated surface ozone, particularly in the Northern Hemisphere.

Finally, we performed a series of model simulations including novel reactions of peroxy radicals with OH. These reactions have been recently confirmed (Bossolasco et al., 2014, Assaf et al., 2017) after being postulated to be potentially important in





the marine boundary layer (Archibald et al., 2009). Using the UM-UKCA model, we have calculated that while these processes cannot account for the missing OH reactivity, they have important implications for the troposphere. Model runs including the reaction of the simplest peroxy radical, $CH_3O_2$, with OH indicated that this process is a major sink of peroxy radicals (with $[CH_3O_2]$ reduced by a third) and an important source of $HO_2$ radicals, the abundance of which increased by up to 12 % over

remote oceans. These runs also show that, with the current understanding of the kinetics and product branching of this process, reaction of $CH_3O_2$ with OH cannot be a major source of atmospheric methanol. As information on the kinetics and products of these reactions become available from laboratory studies, and given the impact they have on tropospheric radical species, we recommend their inclusion in atmospheric models.

There remain a number of challenges in understanding the chemistry of hydroxyl radicals in the atmosphere. We have shown using the UM-UKCA model that accounting for potential new sinks of OH and including a representation of the observed missing OH reactivity in the model has relatively negligible impact on important long-lived atmospheric trace gases. However, we conclude that further studies are needed to identify the source and nature of the observed missing OH reactivity in particular to understand if it acts as a net OH sink (as we have included) or of it couples with other radical propagation cycles and so

feeds back on OH itself.

**Acknowledgements**

We thank the European Research Council for funding through the Atmospheric Chemistry-Climate Interactions (ACCI) project, project number 267760. We would also like to acknowledge the UKCA team at the UK Met Office for their help and support. This work used the ARCHER UK National Supercomputing Service (http://www.archer.ac.uk). ATA thanks the

Kundert Walters Trust and NERC through NE/M00273X/1.

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
