# Peer review of "Global modelling of the total OH reactivity: investigations on the "missing" OH sink and its atmospheric implications"

_Atmospheric Chemistry and Physics, 2018_

## Referee Comment (RC1) · Anonymous Referee #2 · 1 Feb 2018

Review of "Global modelling of the total OH reactivity: investigations on the "missing" OH sink and its atmospheric implications" by Valerio Ferracci and colleagues.

This is a very interesting study that makes a significant contribution to the field. Increasingly, field experimental campaigns include OH reactivity measurements, which provide an important constraint to our understanding of VOC emissions and their atmospheric oxidation processes. Ferracci et al. introduced a hypothetical sink in their state-of-the-art global atmospheric chemistry transport model to study "missing" OH reactivity, i.e. the reactivity that could not be modelled in comparison to field data.

There is one drawback that I would like to see discussed before recommending publication. For the impacts on OH and O3 in section 4.2 it was assumed that the hypothetical emissions of molecule X, probably representing biogenic VOCs, do not recycle OH through their oxidation products (OVOCs). They are assumed to be a simple OH sink without any further chemistry, which is a rather strong simplification. There is growing evidence that biogenic VOCs are unlikely to be ultimate OH sinks and that OH recycling is ubiquitous. Although mentioned on p. 18 and 20, this aspect needs some discussion in view of the interpretation of atmospheric chemistry impacts, notably in the abstract and conclusion section.

Other than that, I recommend publication in ACP with minor revisions.

Minor comments: -Please define missing reactivity more clearly. Is it missing in the sense that accompanying VOC measurements do not account for all reactivity, or missing in the model. Please make the distinction. -p10 bottom/p.11 top: It would be helpful to compare the model calculated OH with some of the published OH measurements in the Amazon. Often, isoprene chemistry mechanisms severely underestimate OH.

**ACPD**

---

## Referee Comment (RC2) · Anonymous Referee #3 · 1 Feb 2018

The article entitled "Global modelling of the total OH reactivity: investigations on the "missing" OH sink and its atmospheric implications" concerns first the modelling of the OH reactivity at the global scale with the model UM-UKCA (base case) and the comparison with measured total OH reactivity in various environments (27 field measurement used for the comparison). The model reproduces well the measured OH reactivity (within 20%) for 12 campaigns, underestimate it for 14 cases and overestimate it for one case. An individual analysis of the differences between the modelled and calculated reactivity per species reacting with OH on 11 categories highlights the main source of underestimation by the model: the NMHCs whereas the overestimation case is due to the isoprene concentration overestimation. This work is of high interest

because there are only a few studies on the oxidant capacity of the atmosphere with global atmospheric modelling and this study is the first one dedicated to the comparison between measured and modelled OH reactivity.

In a second step, an original approach has been used to represent at the global scale the missing reactivity through the addition of a hypothetical molecule X, reacting with OH (but without OH recycling), at various concentrations depending on the missing reactivity measured in the different campaigns. The emission rate of this species has been determined by a multiple linear regression including 15 categories of emission and an iterative procedure to match the missing reactivity observed. The impact of this addition is analysed through the depletion of OH concentration (up to 90% in the eastern Europe for example), the methane and OH lifetime (respectively 2.3 and -2% at the surface) and the ozone change has been used to quantify the impact of this missing reactivity on the OH lifetime at the global scale as well as the impact on the tropospheric methane lifetime in comparison to the base case. This approach allows to represent the missing reactivity at a global scale but the absence of OH recycling due to this species is a limitation to analyse further the consequence of the missing chemistry highlighted by the missing reactivity measured.

In a third step, the UM-UKCA model has been used to study the impact of the reactions of peroxy radicals with OH on OH reactivity which was found to be weak (maximum of 0.12 s-1) and can't be the reason for the missing OH reactivity. In the last part of the article, the impact of the products yields of the reaction of CH3O2+OH has been studied with 3 extra model runs with different branching ratio for the different product channels (producing respectively HO2 or methanol). The branching ratio has been varied for these 2 channels between 1/0 and 0.6/0.4. The conclusion of this work is that, even with the highest branching ratio used for the channel producing methanol (0.4), the additional production of methanol does not explain the underprediction of methanol by the models. Even if of high interest, this part seems to be decoupled from the other parts of the article dedicated to the OH reactivity.

The article is very interesting and the method and the analysis done on the reactivity based on the comparison between measured reactivity and modelled one is well structured. However, I found sometimes difficult to find the information and some improvements in the Figures could help the reader.

**Comments:**

(1) P7: would be interesting to add in Table 2 the total modelled kOH per campaign to identify better the cases described p9 and quantify the under or overestimation mentioned P9 L2-3. A graph with the reactivity modelled vs reactivity calculated and measured would be useful (in the SI?). For example, for the 14 cases with an OH reactivity underestimated significantly by the model, does it correspond to similar missing reactivity with the calculated reactivity? (2) P8: similarly, in Figure 2, it would be useful to add the calculated reactivity including the contribution of the species used in Figure 3 for the calculated and the modelled reactivity. Even if the information is redundant with the Figure 3, the different presentation will help to better identify the different types of environments. The uncertainty varies from one instrument to another one and the same one is used here and in the whole document, please justify this choice. (3) A short discussion on the different techniques used to measure the OH reactivity and the potential impact on the measurements in the different campaigns should be added. (4) P9, Figure 3: the title used for the x axis is not appropriate and confusing, I would change it for "calculated reactivity from measured species" (5) P10, it is mentioned that "it is difficult to establish whether the differences between observed and modelled OH sinks arise from misrepresenting emissions or abundances of the hydroxyl radical itself without comparing modelled and observed [OH], and measurements of the OH concentrations are only available for a small subset of the campaigns considered here." From these campaigns, at least, wouldn't it be possible to provide a "most probable" reason for the difference? (6) P11 L 8: the difference between modelled and measured NO concentration should be modulated considering the uncertainty and the LOD of the instrument. (7) P11 L22: even if the use of additional species and their intermediates

СЗ

provided different results with most of the time a remaining missing reactivity, it would be interesting to see at the global scale the effect of considering that in complement to the use of X which has the disadvantage of being a unique species, with a unique behavior at the global scale. (8) P15: the Figure 5 shows simulated missing reactivity up to 100 s-1, which is a lot higher than the highest one observed. Could it be commented (due to the model or to the measurement, also based on the potential underestimation of the OH reactivity in specific environments)? (9) P16: I do not understand why the OH reactivity calculated at the first step of the iteration is so overestimated. What can be the assumption(s) done on the calculation of the emission rate of X which could explain this disagreement? How does evolve the concentration of X between the first and the last run? (10) P17, L17: wouldn't it be possible to have different runs including different OH recycling to test its influence? (11) P21: It is not clear in the article what is considered to study the impact of the reaction RO2+OH on the OH reactivity, first and second generation peroxy radicals produced in the oxidation of isoprene are mentioned but what about the other products and the products of these reactions? Could you clarify this point? It would be useful also to provide (in the SI ?) a map of the RO2 and also HO2 concentrations. As the reaction of HO2+OH is as fast as the RO2+OH, could you specify if this reaction has been added in the reactivity? (12) P23: as mentioned previously, even if of high interest, the last part of the article, on the impact of the branching ratios for the reaction CH3O2+OH seems to be decoupled from the other parts of the article dedicated to OH reactivity. Indeed, the aim of this study is to determine the impact of the reaction on the HO2, CH3O2 and methanol concentrations but without further analysis on the OH reactivity. A part dedicated to the RO2+OH and the impact of their products on the reactivity, including the different RO2 would fit better in the article than the part only based on the change in the concentration of the products of CH3O2+OH. (13) P25: the change in methanol concentration is only provided in the text whereas it is the main result of this section. It would be useful to provide a figure similar to Figure 11 and 12 but for methanol and for the 3 runs. It would be useful to write in the Legend of Figure 11 and 12 the corresponding run. (14) P25:

the conclusion that the branching ratio needed for this channel (0.8) is not possible seems too extreme because the branching ratio for channel 1 (producing HO2) has been determined only at low pressure.

---

## Referee Comment (RC3) · Anonymous Referee #1 · 6 Feb 2018

This paper presents a study of the global impacts of "missing OH reactivity" in models used to determine the atmospheric oxidising capacity, and investigates the extent to which additional sinks are required to reconcile observations of OH reactivity with model simulations. The authors use an interesting approach to determine the emissions field necessary to improve the agreement between observed and modelled OH reactivity. The impacts of the reaction between OH and CH3O2, and its branching ratio, on budgets for OH, CH3O2 and the global methane lifetime are also discussed.

In general, the paper is well written and will be of interest to the atmospheric science community. However, the discussion would benefit from some additional detail regard-

ing the regions and environments affected most by missing reactivity, and how the OH and HO2 concentrations are affected in the model. The results of this work could also be used to provide some recommendations as to where future measurements of OH reactivity are most needed to give better constraint for modelling of global methane lifetimes and ozone budgets. Minor comments are listed below.

Page 2, line 6: O(1D) production is observed at wavelengths below 340 nm.

Page 3, line 5: Please comment on the location for which 80 % of the total kOH is missing.

Page 4, line 23: Please update the reference to http://iupac.pole-ether.fr/

Page 7-8, Table 2: Does the use of the mean kOH measured over the whole duration of each campaign skew the averages in any way? Do all the field campaigns have similar data coverage throughout the day or throughout the campaign?

Page 8, Figure 2: Is there a reference for the 20 % measurement uncertainty in observed kOH? I would expect this to depend on the specific technique used to measure kOH and the particular instrument configuration.

Page 10, line 1: Please quantify, or avoid, the statement 'reasonably good agreement'.

Page 17, Figure 6: How many iterations are typically required to obtain the emissions field? Is the r2 in the lower panel of Figure 6b skewed by the few points with high reactivity?

---

## Author Comment (AC1) · 16 Apr 2018

This paper presents a study of the global impacts of "missing OH reactivity" in models used to determine the atmospheric oxidising capacity, and investigates the extent to which additional sinks are required to reconcile observations of OH reactivity with model simulations. The authors use an interesting approach to determine the emissions field necessary to improve the agreement between observed and modelled OH reactivity. The impacts of the reaction between OH and $CH_3O_2$, and its branching ratio, on budgets for OH, $CH_3O_2$ and the global methane lifetime are also discussed.

In general, the paper is well written and will be of interest to the atmospheric science community. However, the discussion would benefit from some additional detail regarding the regions and environments affected most by missing reactivity, and how the OH and $HO_2$ concentrations are affected in the model.

We would like to thank the referee for their support of our paper and for their constructive comments. Below are the answer to the reviewer's comments, point by point.

We feel that the regions affected by measured missing reactivity are already described at length in the introduction (where observations of missing reactivity are discussed) and in Section 4.1, where the geographical distribution of the modelled missing reactivity from multiple linear regression is described. We have added a sentence in the conclusions discussing the regions and environments where the multiple linear regression predicts the presence of missing reactivity.

"The multiple linear regression indicated that the areas most affected by the missing reactivity would be tropical remote regions, where biogenic emission dominate, as well as urban regions all over the globe, where anthropogenic emissions are significant. This result agrees with the type of environments in which missing reactivity has been observed."

Effects of X + OH on OH are already shown in Fig 7 and Figs S2 and S3 (now Figures S8 and S9 in the revised manuscript) and discussed in detail in the text. An additional figure (Figure S10) has been included in the Supplementary Information showing that relative changes in $HO_2$ from X + OH largely mirror those in OH, albeit with a somewhat reduced relative magnitude. Changes in OH and $HO_2$ abundances from $CH_3O_2$ + OH are already discussed at length in Section 5.2.

The results of this work could also be used to provide some recommendations as to where future measurements of OH reactivity are most needed to give better constraint for modelling of global methane lifetimes and ozone budgets.

The Southern Hemisphere was notably underrepresented in the sample of measurements analysed in this study. Aircraft measurements of the missing reactivity would also be desirable, especially at tropical and mid-latitude, to better quantify the impact on global methane lifetimes as well as the vertical profile of the missing reactivity, especially above the boundary layer. A comment to address these points was added to the Conclusions.

"Lastly, as observations of the missing reactivity so far are largely limited to ground level measurements in the Northern Hemisphere, further observations in the Southern Hemisphere as well as aircraft measurements both in the boundary layer and the free troposphere would provide additional constraints to the modelled oxidising capacity of the atmosphere."

Minor comments are listed below.

Page 2, line 6: $O(^1D)$ production is observed at wavelengths below 340 nm.

Amended.

Page 3, line 5: Please comment on the location for which 80 % of the total $k_{OH}$ is missing.

Some text was added in the first two paragraphs on page 3 to indicate that the measurements of up to 80% missing reactivity took place in the Amazon, where the impact of large

unidentified biogenic emissions as well as the complex oxidation chemistry of measured and unmeasured BVOCs is largest.

Page 4, line 23: Please update the reference to http://iupac.pole-ether.fr/

Amended.

Page 7-8, Table 2: Does the use of the mean $k_{OH}$ measured over the whole duration of each campaign skew the averages in any way? Do all the field campaigns have similar data coverage throughout the day or throughout the campaign?

There are a number of campaigns with discontinuous temporal coverage that were not included in the analysis presented in the manuscript. These include 4 sets of measurements in Tokyo, Japan, which only reported daytime values of the OH reactivity (Yoshino et al., *Atm. Env.*, 2006; Chatani et al., *Atm. Chem. Phys.*, 2006; Kato et al., *Atm. Env.*, 2011; Yoshino et al., *Atm. Env.*, 2012). Similarly, measurements in Suriname (Sinha et al., *Atm. Chem. Phys.*, 2008) were discarded as they only covered a limited interval of time on a single day.

We believe that, whilst there is always a possibility that the measurement temporal coverage in a field campaign is not uniform or continuous, the choice of campaigns that spanned many weeks (which was the case for the vast majority of the datasets considered for the missing reactivity calculations) does at least minimise that risk.

Page 8, Figure 2: Is there a reference for the 20 % measurement uncertainty in observed $k_{OH}$? I would expect this to depend on the specific technique used to measure $k_{OH}$ and the particular instrument configuration.

The reviewer is correct and the error bars in Figure 2 (and Figure 4) have been amended to reflect the different uncertainties of the instruments used in each campaign.

Page 10, line 1: Please quantify, or avoid, the statement 'reasonably good agreement'.

The line has been changed to better quantify the agreement between model and observations in Figure 3.

"Overall more than half (53 %) of the reactivities calculated from modelled sinks agree with observations within a factor of two, and the vast majority (88 %) within a factor of ten."

Page 17, Figure 6: How many iterations are typically required to obtain the emissions field? Is the $R^2$ in the lower panel of Figure 6b skewed by the few points with high reactivity?

The work described in the manuscript involved 10 iterations of the routine described in figure 6.

Exclusion of the points with highest reactivity ($> 50$ s$^{-1}$) from the fit in the lower panel of Figure 6b still returns an $R^2$ value $> 0.95$ (0.9515 to be precise, *cf*. 0.9588 obtained with the full dataset). It is necessary to remove all points of reactivity higher than 40 s$^{-1}$ to get an $R^2$ value $< 0.95$ (0.9489). It has to be noted that we use 0.95 as an arbitrary threshold for the $R^2$ value. Moreover, progressive removal of points of high reactivity indicates that these high values are not skewing the $R^2$ value significantly: a fit considering only values lower than 30 s$^{-1}$ still returns an $R^2$ of 0.943 (a very modest 1.6% decrease compared to the full data set).

---

## Author Comment (AC2) · 16 Apr 2018

Review of "Global modelling of the total OH reactivity: investigations on the "missing" OH sink and its atmospheric implications" by Valerio Ferracci and colleagues. This is a very interesting study that makes a significant contribution to the field. Increasingly, field experimental campaigns include OH reactivity measurements, which provide an important constraint to our understanding of VOC emissions and their atmospheric oxidation processes. Ferracci et al. introduced a hypothetical sink in their state-of-the-art global atmospheric chemistry transport model to study "missing" OH reactivity, i.e. the reactivity that could not be modelled in comparison to field data. There is one drawback that I would like to see discussed before recommending publication. For the impacts on OH and $O_3$ in section 4.2 it was assumed that the hypothetical emissions of molecule X, probably representing biogenic VOCs, do not recycle OH through their oxidation products (OVOCs). They are assumed to be a simple OH sink without any further chemistry, which is a rather strong simplification. There is growing evidence that biogenic VOCs are unlikely to be ultimate OH sinks and that OH recycling is ubiquitous. Although mentioned on p. 18 and 20, this aspect needs some discussion in view of the interpretation of atmospheric chemistry impacts, notably in the abstract and conclusion section.

We thank the reviewer for their kind words on the manuscript and for their valuable feedback. We understand that the absence of OH recycling constitutes a limitation of this study. However, as also explained at the beginning of Section 4.2, we feel that there are too few constraints to even attempt modelling of OH recycling following reaction 3. In the most optimistic view, it would involve running different recycling scenarios, with each scenario requiring to re-run the iterative routine to determine the emissions of X as the recycling of OH would ultimately

perturb steady state [X] and ultimately $k_3$[X], *i.e.* the modelled missing reactivity. We conclude that the work described in the manuscript provides an upper limit on the effects of the missing reactivity on the oxidising capacity of the atmosphere. We have added a statement to address this in both the abstract and conclusions.

"As no OH recycling was introduced following the initial oxidation of X, these results can be interpreted as an upper limit of the effects of the missing reactivity on the oxidative capacity of the troposphere." and "It has to be noted that, as no OH recycling was introduced following the initial oxidation of X, these results should be interpreted as an upper limit of the effects of the missing reactivity on the oxidative capacity of the troposphere."

Other than that, I recommend publication in ACP with minor revisions.

Minor comments:

-Please define missing reactivity more clearly. Is it missing in the sense that accompanying VOC measurements do not account for all reactivity, or missing in the model. Please make the distinction.

We have added a clearer definition of missing reactivity after Figure 2 in Section 3.

"The total observed $k_{OH}$ in Figure 2 is made up of contributions from the measured OH sinks, from modelled intermediates (only available for some of the field campaigns presented here) and from reactivity that is unaccounted for by known OH sinks, *i.e.* the missing reactivity. "

-p10 bottom/p.11 top: It would be helpful to compare the model calculated OH with some of the published OH measurements in the Amazon. Often, isoprene chemistry mechanisms severely underestimate OH.

The reviewer makes a very valid point. The OH measurements from Liu et al. (Liu et al., *PNAS* 2016) and from the GABRIEL campaign (Martinez et al., *Atm. Chem. Phys.*, 2010) were compared with the model output. It has to be noted that Liu et al. only measured OH for ~7 hours on a single day, and that the GABRIEL campaign in Suriname consisted of airborne measurements. The model underestimated OH by almost a factor of 4 in both cases, indicating that there is a strong possibility that the high concentrations of modelled VOCs for the ATTO site are caused by underpredicted OH. This is consistent with the inverse relationship between OH reactivity and OH concentrations as shown in Figure R1 below.

[Figure]

**Figure R1**: Scatter plot of total OH reactivity against OH concentration. Observations from field studies (in blue) and values from the UM-UKCA model for the same locations (in red) are shown. Also shown is the model output for the surface (grey points) to highlight the inverse relationship between $k_{OH}$ and [OH].

The text has been modified to account for this.

"Figure 3 also offers an explanation for the instances in which the model significantly over predicted $k_{OH}$. For example, the abundance of isoprene measured during the wet season of the ATTO campaign in the Amazon (~1 ± 0.1 ppbv, or nmol/mol, in March 2013) was more than an order of magnitude lower than that predicted by the model for the same time of the year (~14.6 ppbv).  As discussed above, this might arise from either overestimated isoprene emissions or from underestimated OH abundances in the model.  As OH concentrations were not measured during the ATTO campaign, a direct comparison of modelled and observed [OH] is not possible. However [OH] measurements from campaigns carried out in neighbouring parts of the Amazon (Liu et al., 2016) and in the Suriname rainforest (Martinez et al., 2010) might help address this point. Indeed the model underestimates [OH] by almost a factor of four on average in both cases, although it is worth noting that [OH] measurements from Liu et al. (2016) only cover ~7 hours on a single day, while the GABRIEL campaign in Suriname consisted of airborne measurements, and only the OH data for the boundary layer were considered for comparison with the model. It may also be indicative of underrepresented [OH]

in model that the abundance of other short-lived OH sinks in the ATTO campaign is also overestimated by the model; notably, the observed concentration of monoterpenes (reported to be below the detection limit of the PTR-MS used by Nölscher and co-workers, and here approximated to 0.01 ppbv) was much lower than in the model (2.2 ppbv). Underrepresented OH in the model might arise from underestimating the secondary OH originating from the oxidation of large organics (e.g., isoprene and monoterpenes, as described in Archibald et al., 2010). In this specific instance the model also underestimated the concentration of NO (34 pptv, or pmol/mol, vs the observed ~1 ± 0.05 ppbv), which might have limited the production of secondary OH via the reaction of $HO_2$ with NO relative to observations."

---

## Author Comment (AC3) · 16 Apr 2018

The article entitled "Global modelling of the total OH reactivity: investigations on the "missing" OH sink and its atmospheric implications" concerns first the modelling of the OH reactivity at the global scale with the model UM-UKCA (base case) and the comparison with measured total OH reactivity in various environments (27 field measurement used for the comparison). The model reproduces well the measured OH reactivity (within 20%) for 12 campaigns, underestimate it for 14 cases and overestimate it for one case. An individual analysis of the differences between the modelled and calculated reactivity per species reacting with OH on 11 categories highlights the main source of underestimation by the model: the NMHCs whereas the overestimation case is due to the isoprene concentration overestimation. This work is of high interest because there are only a few studies on the oxidant capacity of the atmosphere with global atmospheric modelling and this study is the first one dedicated to the comparison between measured and modelled OH reactivity.

In a second step, an original approach has been used to represent at the global scale the missing reactivity through the addition of a hypothetical molecule X, reacting with OH (but without OH recycling), at various concentrations depending on the missing reactivity measured in the different campaigns. The emission rate of this species has been determined by a multiple linear regression including 15 categories of emission and an iterative procedure to match the missing reactivity observed. The impact of this addition is analysed through the depletion of OH concentration (up to 90% in the eastern Europe for example), the methane and OH lifetime (respectively 2.3 and −2% at the surface) and the ozone change has been used to quantify the impact of this missing reactivity on the OH lifetime at the global scale as well as the impact on the tropospheric methane lifetime in comparison to the base case. This approach allows to

represent the missing reactivity at a global scale but the absence of OH recycling due to this species is a limitation to analyse further the consequence of the missing chemistry highlighted by the missing reactivity measured.

In a third step, the UM-UKCA model has been used to study the impact of the reactions of peroxy radicals with OH on OH reactivity which was found to be weak (maximum of $0.12 \text{ s}^{-1}$) and can't be the reason for the missing OH reactivity. In the last part of the article, the impact of the products yields of the reaction of $CH_3O_2 + OH$ has been studied with 3 extra model runs with different branching ratio for the different product channels (producing respectively HO2 or methanol). The branching ratio has been varied for these 2 channels between 1/0 and 0.6/0.4. The conclusion of this work is that, even with the highest branching ratio used for the channel producing methanol (0.4), the additional production of methanol does not explain the under-prediction of methanol by the models. Even if of high interest, this part seems to be decoupled from the other parts of the article dedicated to the OH reactivity. The article is very interesting and the method and the analysis done on the reactivity based on the comparison between measured reactivity and modelled one is well structured. However, I found sometimes difficult to find the information and some improvements in the Figures could help the reader.

We thank the referee for their kind support of the manuscript and for their constructive comments. Below are the answer to the reviewer's comments.

Comments:

(1) P7: would be interesting to add in Table 2 the total modelled $k_{OH}$ per campaign to identify better the cases described p9 and quantify the under or overestimation mentioned P9 L2-3. A graph with the reactivity modelled *vs* reactivity calculated and measured would be useful (in the SI?). For example, for the 14 cases with an OH reactivity underestimated significantly by the model, does it correspond to similar missing reactivity with the calculated reactivity?

An additional column for the total modelled $k_{OH}$ was added to Table 2.

Plots of modelled reactivity *vs* observed and calculated reactivity were added in the Supplementary Material. Generally, the modelled reactivity is in better agreement with the calculated reactivity than the observed reactivity, even though the model still underestimates the calculated reactivity in urban and suburban environments. This can be mainly accounted for in terms of the reactivity arising from NMHCs, as explained in the text.

As Figure 2 was changed to address the following comment, the discussion on the agreement between modelled and observed reactivity was amended to reflect the new information provided by the new version of Figure 2.

 (2) P8: similarly, in Figure 2, it would be useful to add the calculated reactivity including the contribution of the species used in Figure 3 for the calculated and the modelled reactivity. Even if the information is redundant with the Figure 3, the different presentation will help to better identify the different types of environments. The uncertainty varies from one instrument to another one and the same one is used here and in the whole document, please justify this choice.

We changed Figure 2 to show the calculated reactivity from measured species as well as the reactivity from reaction intermediates (when available) and the missing reactivity. We have also included a version of Figure 2 with the speciation of the reactivity (hence incorporating the information in Figure 3) in the Supplementary Material as Figure S5. As the reviewer suggested, this figure offers an effective way to distinguish the different environments: urban sites are dominated by $NO_x$ and NMHCs; remote sites by biogenic VOCs (mainly isoprene and its oxidation products), with suburban environments sitting somewhere in between the previous two.  The text between Figures 2 and 3 (as well as the caption of Figure 2) has been amended to reflect this change.

The error bars in Figure 2 (and also in Figure 4) have been amended to reflect the different uncertainties of the instruments used in each campaign.

 (3) A short discussion on the different techniques used to measure the OH reactivity and the potential impact on the measurements in the different campaigns should be added.

We have added some additional text on page 2 to describe the main techniques used to measure $k_{OH}$ in the field, however we feel that a too detailed account would detract from the main objectives of the manuscript. The reader is referred to two recent works (Yang et al., 2016 and Fuchs et al., 2017) in which the experimental techniques are extensively reviewed.

(4) P9, Figure 3: the title used for the x axis is not appropriate and confusing, I would change it for "calculated reactivity from measured species"

Amended (the $y$-axis label was also amended).

(5) P10, it is mentioned that "it is difficult to establish whether the differences between observed and modelled OH sinks arise from misrepresenting emissions or abundances of the hydroxyl radical itself without comparing modelled and observed [OH], and measurements of the OH concentrations are only available for a small subset of the campaigns considered here." From these campaigns, at least, wouldn't it be possible to provide a "most probable" reason for the difference?

A correlation plot of modelled *vs* observed [OH] was added in the Supplementary Material for those campaigns that measured OH as well as the total reactivity and the individual OH sinks. In a couple of cases in which the model overestimated [OH] by roughly a factor of six (BEARPEX09 and CABINEX), the modelled isoprene concentrations were approximately a factor of two lower than the observations in both cases. There are no OH measurements available for the campaigns that exhibited the largest disagreement between observations and model (namely, ATTO), but analysis of OH measurements in Amazonia from different campaigns (see response to reviewer 2) indicate that the model might underrepresent OH in these locations, hence resulting in higher isoprene and VOCs than in the observations. This behaviour is illustrated in Figure R1 in the response to reviewer 2. The text on pages 10-12 has been amended accordingly.

"A correlation plot of modelled against observed [OH] is given in the Supplementary Material (Figure S6) for those campaigns that measured OH as well as the total reactivity and the individual OH sinks. In a couple of cases in which the model overestimated [OH] by roughly a factor of six (BEARPEX09 and CABINEX), the modelled isoprene concentrations were approximately a factor of two lower than the observations in both cases."

 (6) P11 L8: the difference between modelled and measured NO concentration should be modulated considering the uncertainty and the LOD of the instrument.

The NO instrument used for the ATTO measurements (Ecophysics chemiluminescence analyser, model CLDTR-780) has a LOD = 0.05 ppb and uncertainty < 5% (as reported by Williams et al., *Atm. Env.*, 2016). Even in the light of the measurement uncertainty, the observed [NO] is considerably larger than in the model. We have added the measurement uncertainty to the text for clarity. We have also added the uncertainty in the isoprene measurement (as reported by Yanez-Serrano et al., *Atm. Chem. Phys.*, 2015) at the beginning of the same paragraph.

(7) P11 L22: even if the use of additional species and their intermediates provided different results with most of the time a remaining missing reactivity, it would be interesting to see at the global scale the effect of considering that in complement to the use of X which has the disadvantage of being a unique species, with a unique behavior at the global scale.

This approach, while feasible for box models used to interpret the results of the individual field campaigns, would be extremely burdensome for a global model like UM-UKCA. It is still an interesting area, worth investigating in future studies.

(8) P15: the Figure 5 shows simulated missing reactivity up to 100 s$^{-1}$, which is a lot higher than the highest one observed. Could it be commented (due to the model or to the measurement, also based on the potential underestimation of the OH reactivity in specific environments)?

The vast majority of the missing reactivity values plotted in Figure 5 are below 10 s$^{-1}$ (87% of all non-zero entries), and the near totality are below 50 s$^{-1}$ (99.8%).

Values of the missing reactivity higher than 50 s$^{-1}$ are only found in 12 grid cells over the whole globe in DJF and in 15 grid cells JJA in Figure 5. Upon closer inspection, these correspond to areas of high anthropogenic emissions (principally large urban areas), listed in the table below for the JJA plot in Figure 5. For the majority of these entries the missing reactivity is dominated by large contributions from the highly-correlated organic and black carbon emissions from biofuels (OC and BC biofuel emissions in Table 3 in the manuscript). However in a small number of cases (Caracas, Dubai, Kuwait City) the missing reactivity appears dominated by emissions related to the oil refinery industry (propane, formaldehyde).

| Longitude / ° E | Latitude / ° N | Location | Missing reactivity from MLR / s$^{-1}$ |
|---|---|---|---|
| 112.5 | -7.5 | Surabaya (East Java), Indonesia | 50.65 |
| 106.875 | -6.25 | Jakarta, Indonesia | 60.61 |
| 292.5 | 10 | Caracas, Venezuela | 58.27 |
| 292.5 | 11.25 | Caracas, Venezuela | 56.21 |
| 106.875 | 21.25 | Hanoi, Vietnam | 56.37 |
| 90 | 23.75 | Dhaka, Bangladesh | 81.36 |
| 56.25 | 25 | Dubai, UEA | 94.30 |
| 90 | 25 | Brahmaputra river, Bangladesh | 67.12 |
| 48.75 | 28.75 | Kuwait City, Kuwait | 66.17 |
| 48.75 | 30 | Kuwait/Iraq/Iran border, North Persian Gulf | 85.46 |
| 69.375 | 41.25 | Tashkent, Uzbekistan | 60.37 |
| 26.25 | 45 | Bucharest, Romania | 62.71 |
| 28.125 | 47.5 | Chisinau, Moldova | 56.93 |

| 37.5 | 56.25 | Moscow, Russia | 99.99 |
| 30 | 60 | St Petersburg, Russia | 91.78 |

We believe these high values are ultimately a result of the MLR approach used in this work. A paragraph describing the distribution and magnitude of the modelled missing reactivity across the globe has been added to Section 4.1 (after Figure 4 and before Table 3).

"Overall, the modelled missing reactivity obtained from the multiple linear regression had values other than zero in 23 % of the surface grid cells in DJF and in 32 % of the grid cells in JJA. Of all the non-zero values plotted in Figure 5, 57 % are below 1 s$^{-1}$, 77 % below 5 s$^{-1}$, 87 % are below 10 s$^{-1}$ and 99.8 % are below 50 s$^{-1}$. Only a very small number of grid cells have modelled missing reactivities in the range 50-100 s$^{-1}$ (12 grid cells in DJF and 15 in JJA). These regions correspond to areas of high anthropogenic emissions that resulted in large contributions of the strongest predictors (OC and BC biofuels) to the calculated missing reactivity."

(9) P16: I do not understand why the OH reactivity calculated at the first step of the iteration is so overestimated. What can be the assumption(s) done on the calculation of the emission rate of X which could explain this disagreement? How does evolve the concentration of X between the first and the last run?

We believe that the large overestimate in OH reactivity due to X after the first step in the iteration is due to the values of [OH] used in Eq. (2). As stated in the text, the [OH] field is taken from the base run (*i.e.*, in the absence of X). As the introduction of an additional OH sink perturbs the OH field itself (leading to lower OH abundances), the OH field from the base run is itself an overestimate of the OH abundances found when X and OH are in steady state. A sentence describing this was added to the manuscript.

Overall the concentration of X decreases between the first and the last run as shown in Figure R2 below (mean and median [X]). The first iteration shows a very large overestimate, for the reasons described above. The second iteration somewhat over-corrects for the first one and the subsequent ones give more or less a steady mean/median values of [X].

[Figure]

**Figure R2**: Variation of mean (left) and median (right) [X] as a function of iteration number.

(10) P17, L17: wouldn't it be possible to have different runs including different OH recycling to test its influence?

As we mentioned in our reply to Reviewer 2, introducing different OH recycling would involve running the iterative routine to determine the emissions of X in each scenario, as the recycling of OH would ultimately perturb steady state [X] and ultimately $k_3$[X], *i.e.* the modelled missing reactivity. We prefer presenting the work described in the manuscript as an upper limit of the effects of the missing reactivity on the oxidising capacity of the atmosphere. As requested by Reviewer 2, we have re-iterated this in both the abstract and conclusions.

(11) P21: It is not clear in the article what is considered to study the impact of the reaction $RO_2$ + OH on the OH reactivity, first and second generation peroxy radicals produced in the oxidation of isoprene are mentioned but what about the other products and the products of these reactions? Could you clarify this point? It would be useful also to provide (in the SI?) a map of the $RO_2$ and also $HO_2$ concentrations. As the reaction of $HO_2$ + OH is as fast as the $RO_2$+OH, could you specify if this reaction has been added in the reactivity?

A clearer list of the peroxy radicals included in the model was added to the first paragraph of Section 5.1. We have also specified more clearly that the contribution of all $RO_2$ radicals to the total $k_{OH}$ was calculated offline, so that the individual $RO_2$ + OH reactions were not added to the model, as this would require some degree of knowledge of their products. The one exception is of course the reaction of methyl peroxy + OH, which is described in Section 5.2.

Maps of $HO_2$ and total $RO_2$ concentrations were added to the Supplementary Material as Figures S2 and S3 respectively..

The $HO_2$ + OH reaction was included in the main reactivity calculations described in Section 3 of the manuscript. The magnitude of this term is smaller than that calculated for the total $RO_2$, as shown below in Figure R3 (in s$^{-1}$, using the same scale as Figure 10a and 10b).

[Figure]

**Figure R3**: Annual mean reactivity from the $HO_2$ + OH reaction in s$^{-1}$.

(12) P23: as mentioned previously, even if of high interest, the last part of the article, on the impact of the branching ratios for the reaction $CH_3O_2$ + OH seems to be decoupled from the other parts of the article dedicated to OH reactivity. Indeed, the aim of this study is to determine the impact of the reaction on the $HO_2$, $CH_3O_2$ and methanol concentrations but without further analysis on the OH reactivity. A part dedicated to the $RO_2$ + OH and the impact of their products on the reactivity, including the different $RO_2$ would fit better in the article than the part only based on the change in the concentration of the products of $CH_3O_2$ + OH.

A paragraph was added at the end of Section 5.2 to better address how the chemistry of $CH_3O_2$ + OH fits within with the wider theme of OH reactivity; figures have also been added in the Supplementary Material to illustrate the changes in $k_{OH}$ brought about by R4 (Figures S13 and S14).

"As the species affected by R4 ($HO_2$, $CH_3O_2$ and $CH_3OH$) are all OH sinks, changes in their concentrations are accompanied by changes in $k_{OH}$. However these are modest ($< 0.25$ s$^{-1}$) in all the scenarios considered in this work, as shown in Figures S13 and S14 in the Supplementary Material. While the reactions of OH with both $HO_2$ and $CH_3O_2$ have large rate constants ($> 1 \times 10^{-10}$ cm$^3$ molecules$^{-1}$ s$^{-1}$), the general low abundance of these species (of the

order of ~$10^8$ molecules cm$^{-3}$, as shown in Figures S2 and S3) results in small changes to the total $k_{OH}$. On the other hand, while some of the changes in methanol concentrations arising from R4 are significant (with increases up to 400 pptv in run 3, corresponding to ~1 ×$10^{10}$ molecules cm$^{-3}$), the very small rate constant of its reaction with OH (< 1× $10^{-12}$ cm$^3$ molecules$^{-1}$ s$^{-1}$) leads to a small contributions to $k_{OH}$. These results are consistent with the magnitude of the changes in $k_{OH}$ calculated offline for all RO$_2$ radicals in Section 5.1. It remains to be seen if any of the reaction products of more complex RO$_2$ radicals with OH, or their combination, might have a significant impact on $k_{OH}$."

(13) P25: the change in methanol concentration is only provided in the text whereas it is the main result of this section. It would be useful to provide a figure similar to Figure 11 and 12 but for methanol and for the 3 runs. It would be useful to write in the Legend of Figure 11 and 12 the corresponding run.

Agreed. A figure showing the changes in methanol under the three scenario was added (Figure 13).

The corresponding run for both Figures 11 and 12 was added to the captions.

(14) P25: the conclusion that the branching ratio needed for this channel (0.8) is not possible seems too extreme because the branching ratio for channel 1 (producing HO$_2$) has been determined only at low pressure.

Agreed. As theoretical studies seem to indicate the presence of a termolecular association channel (Müller et al., *Nat. Commun.*, 2016; but also Liu et al., *Chem. Res. Chin. Univ.*, 2017 for ethyl peroxy + OH) leading to the formation of CH$_3$OOOH (which could potentially decompose to CH$_3$OH and O$_2$), we have added a sentence to highlight the need for further characterisation of the branching at ambient pressure.

"However it has to be noted that so far the product branching of R4 has only been measured at low pressure (50 Torr) by Assaf et al. (2017b). Calculations by Müller et al. (2016) suggest the presence of an association channel leading to the formation of a trioxide (CH$_3$OOOH) species, which might potentially decompose to methanol and molecular oxygen. As the stabilisation of association products is generally a pressure-dependent process, it is very important that future studies address the branching of R4 at ambient pressure."

---

## Author Comment (AC4) · 16 Apr 2018

**Global modelling of the total OH reactivity: investigations on the "missing" OH sink and its atmospheric implications**

Valerio Ferracci[1,a], Ines Heimann[1], N. Luke Abraham[1,2], John A. Pyle[1,2] and Alexander T. Archibald[1,2]

[1]Centre for Atmospheric Science, Department of Chemistry, University of Cambridge, Lensfield Road, CB2 1EW, UK
[2]National Centre for Atmospheric Science, University of Cambridge, Cambridge, UK
[a]now at: Centre for Environmental and Agricultural Informatics, Cranfield University, College Road, MK43 0AL, UK

*Correspondence to*: Valerio Ferracci (v.ferracci@cranfield.ac.uk)

**Abstract.** The hydroxyl radical (OH) plays a crucial role in the chemistry of the atmosphere as it initiates the removal of most trace gases. A number of field campaigns have observed the presence of a "missing" OH sink in a variety of regions across the planet. Comparison of direct measurements of the OH loss frequency, also known as total OH reactivity ($k_{OH}$), with the sum of individual known OH sinks (obtained *via* the simultaneous detection of species such as volatile organic compounds and nitrogen oxides) indicates that, in some cases, up to 80 % of $k_{OH}$ is unaccounted for. In this work, the UM-UKCA chemistry-climate model was used to investigate the wider implications of the missing reactivity on the oxidising capacity of the atmosphere. Simulations of the present-day atmosphere were performed and the model was evaluated against an array of field measurements to verify that the known OH sinks were reproduced well, with a resulting good agreement found for most species. Following this, an additional sink was introduced to simulate the missing OH reactivity as an emission of a hypothetical molecule, X, which undergoes rapid reaction with OH. The magnitude and spatial distribution of this sink were underpinned by observations of the missing reactivity. Model runs showed that the missing reactivity accounted for on average 6 % of the total OH loss flux at the surface, and up to 50 % in regions where emissions of the additional sink were high. The lifetime of the hydroxyl radical was reduced by 3 % in the boundary layer, while tropospheric methane lifetime increased by 2 % when the additional OH sink was included. As no OH recycling was introduced following the initial oxidation of X, these results can be interpreted as an upper limit of the effects of the missing reactivity on the oxidative capacity of the troposphere. The UM-UKCA simulations also allowed us to establish the atmospheric implications of the newly characterised reactions of peroxy radicals ($RO_2$) with OH. While the effects of this chemistry on $k_{OH}$ were minor, the reaction of the simplest peroxy radical, $CH_3O_2$, with OH was found to be a major sink for $CH_3O_2$ and source of $HO_2$ over remote regions at the surface and in the free troposphere. Inclusion of this reaction in the model increased tropospheric methane lifetime by up to 3 %, depending on its product branching. Simulations based on the latest kinetic and product information showed that this reaction cannot reconcile models with observations of atmospheric methanol, in contrast to recent suggestions.

**1 Introduction**

The removal of the vast majority of trace gases emitted into the atmosphere is initiated by reaction with the hydroxyl radical, OH. OH is primarily formed following the reaction of excited oxygen atoms, $O(^1D)$, originating from the photolysis of ozone at short wavelengths, with water:

$O_3 + h\nu \ (\lambda < \text{310 } 340 \text{ nm}) \ \rightarrow \qquad O(^1D) + O_2,$ \hfill (R1)

[revised manuscript text omitted]

---

## Author Comment (AC6) · 16 Apr 2018

The comment was uploaded in the form of a supplement:
https://www.atmos-chem-phys-discuss.net/acp-2018-12/acp-2018-12-AC6-
supplement.pdf